# Accelerating reliable multiscale quantum refinement of protein–drug systems enabled by machine learning

Zeyin Yan [1], Dacong Wei[1], Xin Li[1] & Lung Wa Chung [1] ✉

Biomacromolecule structures are essential for drug development and biocatalysis. Quantum refinement (QR) methods, which employ reliable quantum mechanics (QM) methods in crystallographic refinement, showed promise in improving the structural quality or even correcting the structure of biomacromolecules. However, vast computational costs and complex quantum mechanics/molecular mechanics (QM/MM) setups limit QR applications. Here we incorporate robust machine learning potentials (MLPs) in multiscale ONIOM(QM:MM) schemes to describe the core parts (e.g., drugs/inhibitors), replacing the expensive QM method. Additionally, two levels of MLPs are combined for the first time to overcome MLP limitations. Our unique MLPs +ONIOM-based QR methods achieve QM-level accuracy with significantly higher efficiency. Furthermore, our refinements provide computational evidence for the existence of bonded and nonbonded forms of the Food and Drug Administration (FDA)-approved drug nirmatrelvir in one SARS-CoV-2 main protease structure. This study highlights that powerful MLPs accelerate QRs for reliable protein–drug complexes, promote broader QR applications and provide more atomistic insights into drug development.

Accurate atomic structures of biomacromolecules are vital for molecular property prediction, binding pose estimation, as well as understanding ligand binding site recognition and biocatalysis. Additionally, this structural information plays an indispensable role in the rational development and design of new drugs with high potency and selectivity that specifically target the binding site[1–3]. In this regard, X-ray diffraction (XRD) has long been one of the most powerful methods to determine the atomic structures of many biomacromolecules. Structural determination often relies on standard X-ray crystallographic refinement methods, in which the molecular mechanics (MM) force field is combined with experimental (XRD) data to derive reasonable chemical structures[4,5]. However, the development of force fields (using limited parameters) to give reliable structures of diversified drug molecules has long been challenging due to the enormous variety of chemical space (with many element–element combinations) and complex electronic effects (such as conjugation/delocalization)[6].

Recent breakthroughs in the development of various artificial intelligence (AI) methods (e.g., AlphaFold or RoseTTAFold) can predict impressively reasonable atomic structures of some proteins[7,8]. Unfortunately, these AI-based methods (such as AlphaFold) still cannot easily predict reliable biological structures containing cofactors or drugs/inhibitors, due to scarce experimental structural data[9,10].

On the other hand, the development of quantum refinement (QR, pioneered by Ryde)[11], which replaces the MM method with more accurate quantum mechanics (QM) methods, can overcome the challenges of reliably describing various drug structures[12]. In fact, the QR method has been successfully applied to some protein–drug/inhibitor systems, such as acetylcholinesterase with the anti-Alzheimer drug donepezil[13] and serine proteases with benzamidinium-based inhibitors[14]. These successful applications of QR to biomacromolecules have demonstrated many promising results in improving the structural quality and even giving the correct structures[15–19]. Moreover,

[1]Shenzhen Grubbs Institute, Department of Chemistry and Guangdong Provincial Key Laboratory of Catalysis, Southern University of Science and Technology, Shenzhen 518055, China. ✉e-mail: oscarchung@sustech.edu.cn

recent developments in QR combined with multiscale[11,13,15–22], linear-scale QM[23,24], fragmentation[25–28] and quantum-embedding[22] methods by different groups further boost or improve the refinement process. Nevertheless, compared to the fast MM methods usually used in crystallographic refinements, the much higher computational costs and complex setup of QM/MM systems hinder the broad applications of QR to many biological systems.

Recently, active AI method development (such as machine learning potentials (MLPs)) has emerged as a promising alternative that can quickly predict energy and related gradients after training QM energies and gradients with very large datasets of various atomic configurations. MLPs are remarkable for their robustness and for accuracy comparable to that of high-level ab initio methods used to generate training data[29–32]. ANI series using deep neural networks (DNNs) are powerful and successful MLPs (achieving density functional theory (DFT) or coupled-cluster (CC) accuracy) that have attracted much attention in recent years[33–35] due to their flexibility and transferability to a wide range of molecules and systems. However, MLPs are still less transferable than QM methods and are generally limited to closed-shell, neutral organic compounds. The Dral group elegantly developed a general-purpose artificial intelligence–quantum mechanical method 1 (AIQM1) to approach coupled-cluster (CC) accuracy[36] by combining the Δ-machine learning (ML) strategy[37] and

semi-empirical (SE) method[38], using ANI-type neural network (and datasets) potentials and D4-dispersion[39] as corrections. The AIQM1 method displayed good accuracy even for challenging systems (such as ions and excited states). Additionally, several groups adopted ML to replace or correct the expensive QM method in ab initio QM/MM methods[40–42] to enhance accuracy and reduce the high computational cost of QM/MM molecular dynamics simulations[43–48]. Inspired by these encouraging results in new MLPs, we envision that MLPs will introduce a new opportunity to develop a faster and more accurate QR method[49,50], since high-level CC methods are rarely used in QR applications due to their prohibitive computational costs[22].

In this study, MLPs are incorporated for the first time as the high layer (Fig. 1a) in the multiscale QR method[22], in which the expensive QM method(s) is replaced by the much faster ANI or AIQM1 method. Due to the high dependence on training data, the MLPs are limited to few elements (e.g.,: AIQM1: C, H, O, N; ANI-2x: C, H, O, N, F, Cl, S) or specific systems (ANI: neutral systems). To apply refinement of drug/inhibitor molecules containing more elements and to overcome such limitation while maintaining the highest accuracy on the core drug/inhibitor structures, two different levels (CC- and DFT-quality) of MLPs (denoted as MLP-CC and MLP-DFT) were further combined for the first time through an extrapolative Our own N-layered Integrated molecular Orbital and molecular Mechanics (ONIOM) approach and introduced

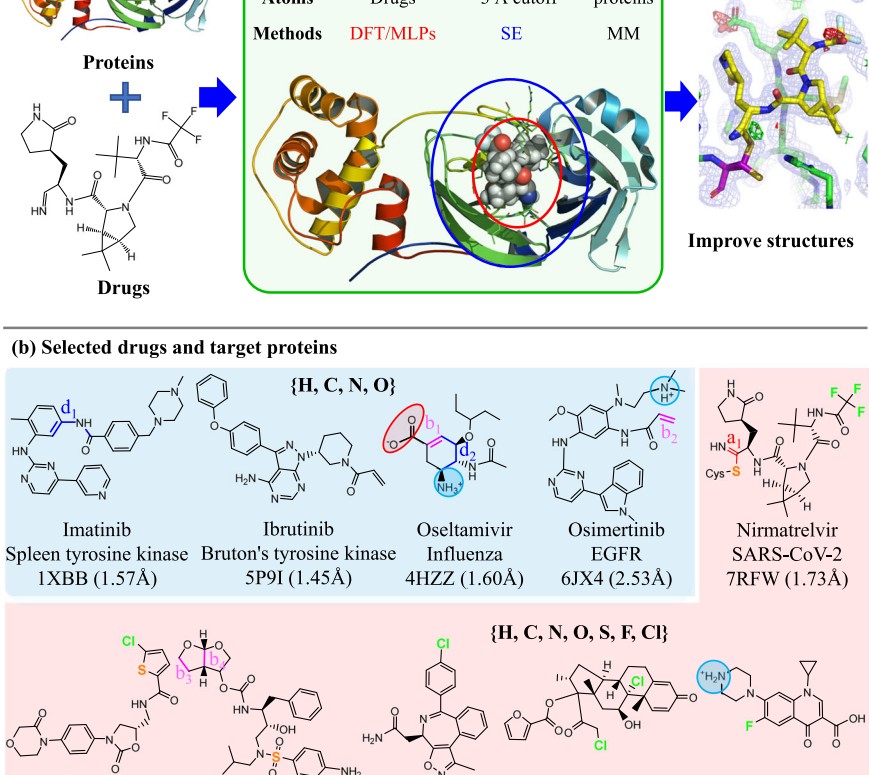

**(a) Multiscale quantum refinements with MLPs**

| ONIOM | High | Medium | Low |
|---|---|---|---|
| Atoms | Drugs | ~ 3 Å cutoff | proteins |
| Methods | DFT/MLPs | SE | MM |

Proteins + Drugs → Improve structures

**(b) Selected drugs and target proteins**

{H, C, N, O}

Imatinib
Spleen tyrosine kinase
1XBB (1.57Å)

Ibrutinib
Bruton's tyrosine kinase
5P9I (1.45Å)

Oseltamivir
Influenza
4HZZ (1.60Å)

Osimertinib
EGFR
6JX4 (2.53Å)

Nirmatrelvir
SARS-CoV-2
7RFW (1.73Å)

{H, C, N, O, S, F, Cl}

Rivaroxaban
Factor Xa
2W26 (2.08Å)

Darunavir
HIV-1
5KR1 (1.60Å)

CPI-0610
Bromodomain
5HLS (2.18Å)

Mometasone
Hormone receptor
4P6W (1.95Å)

Ciprofloxacin
Salmonella typhi OmpF
4KRA (3.32Å)

**Fig. 1 | Schematic workflow and selected protein–drug/inhibitor systems. a** Schematic workflow for our multiscale quantum refinements (QRs) on protein–drug/inhibitor systems combined with machine learning potentials (MLPs). The multiscale method combined with MLPs was highlighted. **b** Chemical structures of ten selected drugs/inhibitors (QR10 dataset; the negative and positive charged groups were marked by red oval and blue circle, respectively). The drug/inhibitor that contains only H, C, N, and O elements were highlighted using blue background color, whereas the drug/inhibitor that contains also S, F, and Cl elements were highlighted using red background color. The drug/inhibitor name, related target protein name, and PDB ID are included beneath the image. Those for the other 40 drugs/inhibitors evaluated in our study are also given in Supplementary Figs. 1–4.

in our ONIOM QR schemes (i.e. ONIOM2(MLP:MM), ONIOM3(MLP:SE:MM), especially the unprecedented ONIOM3(MLP-CC:MLP-DFT:MM) and ONIOM4(MLP-CC:MLP-DFT:SE:MM) schemes, see Table 1). The geometries of 50 different protein–drug/inhibitor systems were then refined and evaluated by various above-mentioned ONIOM-based QR schemes (Fig. 1, Supplementary Figs. 1–4, and Table 1), along with X-ray experimental data. Our results demonstrate that these MLPs+ONIOM-based QR schemes can successfully reach QM-level accuracy with much higher efficiency, which should promote much broader applications of QR methods and be helpful for drug development/design. Moreover, our refinements offer computational evidence for coexisting the bonded and nonbonded forms of the FDA-approved drug nirmatrelvir in one crystal structure of SARS-CoV-2 main protease ($M^{Pro}$), which should be helpful for designing better SARS-CoV-2 drugs.

## Results

### Drug/Inhibitor structures in the gas phase

To assess the performance of a few MLPs, geometry optimization of our selected 50 drugs/inhibitors (QR50 dataset, Figs. 1b, 2 and Supplementary Figs. 1–4; the optimized geometries are provided in Supplementary Data) in the gas phase was first performed using the ωB97X-D/6-31G(d), AIQM1, ANI-1ccx, ANI-2x, ANI-1x, and second-generation Geometry, Frequency, Noncovalent, eXtended Tight Binding (GFN2-xTB) methods. Compared to the reliable results optimized by the (QM) ωB97X-D method, our structural analysis shows that median absolute deviation (MAD) in the bond distances, angles and rotatable dihedrals of the drug/inhibitor molecules by the MLPs (AIQM1, ANI-2x) and SE (GFN2-xTB) methods vary by 0.005–0.008 Å, 0.6–0.9° and 11.2–16.1°, respectively (Table 2). In general, all MLPs led to similar drug/inhibitor structures to those optimized by the DFT method with all the median (white dots) and the highest deviation distribution (maximum width) locations close to zero (Fig. 2). However, the GFN2-xTB method underestimated bond distances, with a median bond deviation distribution close to −0.01 Å.

To further evaluate the accuracy of the MLPs methods for more different functional groups[51], a new computational benchmark dataset taken from PDBbind v2020 (denoted as PB20-QM, https://github.com/oscarchung-lab/PB20-QM) containing 12,963 drug/inhibitor molecules as well as its two smaller sub-datasets (PB20-QM-8 k: 8776 molecules containing, C, H, O, N, F, S, and/or Cl elements; and PB20-QM-3 k: 3156 molecules mainly containing, C, H, O, and/or N elements) were also set up for geometry optimization by the different methods (Table 2). Compared to the reliable ωB97X-D method, our computational results show that MAD in the bond distances, angles and rotatable dihedrals of the drug/inhibitor molecules by the MLPs (AIQM1, ANI-2x) and SE (GFN2-xTB) methods vary by 0.003–0.006 Å, 0.4–0.6°, and 26.7–32.0°, respectively (Table 2).

In addition, the drug/inhibitor molecules containing charged group(s) were generally found to have larger structural deviations (e.g., MAD) when optimized by all MLPs and SE methods than the neutral drug/inhibitor cases (Table 2), except the dihedrals in PB20-QM-3k possibly due to its scarce systems. For the neutral cases, these MLPs showed higher accuracy than GFN2-xTB with smaller structural deviation in bond (ΔMAD: 0.002–0.004 Å) and angle (ΔMAD: 0.1–0.2°) compared to the DFT method. The dihedral deviations for the AIQM1 and GFN2-xTB methods are generally comparable and smaller than those for MLP ANI-2x. Even the ANI-series MLPs were not primarily designed for charged systems, the ANI-2x method still showed a small structural deviation (MAD: <0.009 Å (bond), <0.2° (angle)) for those containing charged group(s). In comparison, the CC-level MLP AIQM1 method gives superior results to the ANI-series MLPs. Therefore, these MLPs, particularly the CC-quality AIQM1 method, can give reliable structures for drug/inhibitor systems at much lower computational costs. Moreover, apart from the charge effect, the structural flexibility of the drug/inhibitor molecules is another key important factor in the larger RMSDs in the gas phase (Supplementary Table 36).

**Table 1 | Computational chemistry methods in our quantum refinements[a]**

|      | Methods |
|------|---------|
| M1   | ONIOM2(DFT:MM) |
| M2   | ONIOM2-EE(DFT:MM) |
| M3   | ONIOM2(ANI-2x:MM) |
| M4   | ONIOM2(ANI-1ccx:MM) |
| M4a  | ONIOM3(ANI-1ccx:ANI-2x:MM)[b] |
| M5   | ONIOM2(AIQM1:MM)[c] |
| M5a  | ONIOM3(AIQM1:ANI-2x:MM)[b] |
| M6   | ONIOM2(SE:MM) |
| M7   | ONIOM3(DFT:SE:MM) |
| M8   | ONIOM3(ANI-2x:SE:MM) |
| M9   | ONIOM3(ANI-1ccx:SE:MM) |
| M9a  | ONIOM4(ANI-1ccx:ANI-2x:SE:MM)[b] |
| M10  | ONIOM3(AIQM1:SE:MM)[c] |
| M10a | ONIOM4(AIQM1:ANI-2x:SE:MM)[b] |
| M6R  | ONIOM2(SE:MM)[d] |

[a]ωB97X-D/6-31 G(d) as the density functional theory (DFT) method, ANI-2x (machine learning potentials with DFT accuracy, MLP-DFT), ANI-1ccx (machine learning potentials with coupled-cluster accuracy, MLP-CC) and AIQM1 (MLP-CC) as machine learning potentials (MLPs), Amber ff14SB as the MM method, and GFN2-xTB as the semi-empirical (SE) method.
[b]Two machine learning potentials were used to describe the drug/inhibitor molecules due to the element limitations (C, H, O, N) of ANI-1ccx and AIQM1.
[c]AIQM1 combined with GFN2-xTB methods were used to describe the drug/inhibitor molecules containing P or Br elements due to the element limitations (C, H, O, N) of AIQM1.
[d]The high layer of **M6R** includes the drug/inhibitor molecule and its neighboring residues within 3 Å.

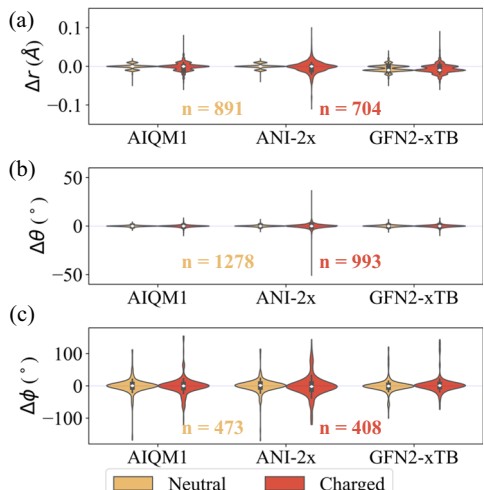

**Fig. 2 | Comparison of the optimized structure in the gas phase.** Violin plots of deviation in (**a**) bond distances (Δr), (**b**) angles (Δθ) and (**c**) rotatable dihedrals (Δφ) for the selected 50 drug/inhibitor structures (QR50 dataset) optimized in the gas phase, using machine learning potentials (MLPs including coupled-cluster accuracy (MLP-CC): AIQM1, and density functional theory accuracy (MLP-DFT): ANI-2x), and the semi-empirical (SE) GFN2-xTB method compared to the (DFT) ωB97X-D/6-31 G(d) method. ONIOM(MLP:ANI-2x) method was used for the molecules containing F, Cl and/or S elements, when AIQM1 was used. ONIOM(MLP:SE) method was used for the molecules containing P and Br elements when AIQM1 or ANI-2x was used. The number of data was given in each plot. The white dots indicate the median values and the inner boxplots indicate the interquartile range. Source data are provided as a Source Data file.

**Table 2 | Comparison of the optimized structures in the gas phase for different datasets**

| | QR50[a] | | | PB2O-QM-3k[a] | | | PB2O-QM-8k[a] | |
|---|---|---|---|---|---|---|---|---|
| | AIQM1[b,c] | ANI-2x[c] | xTB | AIQM1[b,c] | ANI-2x[c] | xTB | ANI-2x | xTB |
| *All systems* | | (50)[d] | | | (3156)[d] | | | (8776)[d] | |
| Bond (Å) | 0.005 | 0.006 | 0.008 | 0.004 | 0.003 | 0.006 | 0.003 | 0.006 |
| Angle (°) | 0.6 | 0.9 | 0.8 | 0.4 | 0.5 | 0.6 | 0.6 | 0.7 |
| Dihedral (°) | 11.6 | 16.1 | 11.2 | 26.7 | 32.0 | 28.0 | 32.6 | 29.0 |
| *Neutral group(s)* | | (24)[d] | | | (3061)[d] | | | (7260)[d] | |
| Bond (Å) | 0.004 | 0.003 | 0.007 | 0.004 | 0.003 | 0.006 | 0.003 | 0.006 |
| Angle (°) | 0.5 | 0.6 | 0.7 | 0.4 | 0.4 | 0.6 | 0.5 | 0.7 |
| Dihedral (°) | 10.6 | 12 | 10.6 | 27.2 | 32.5 | 28.6 | 35.2 | 31.6 |
| *Charged group(s)* | | (26)[d] | | | (95)[d] | | | (1516)[d] | |
| Bond (Å) | 0.006 | 0.009 | 0.010 | 0.005 | 0.004 | 0.005 | 0.004 | 0.006 |
| Angle (°) | 0.7 | 1.3 | 0.9 | 0.6 | 0.8 | 0.8 | 0.8 | 0.8 |
| Dihedral (°) | 12.7 | 21 | 11.9 | 18.2 | 25.0 | 18.5 | 25.9 | 21.5 |

[a]QR50: our selected 50 drugs/inhibitors (Fig. 1b and Supplementary Figs. 1–4); PB2O-QM-3k: the smallest subset dataset of PB2O-QM containing 3156 drugs/inhibitors (3125 molecules containing C, H, O and/or N elements, 15 molecules containing F, Cl and/or S elements, and 16 molecules containing B, P, Se, Br and/or I elements); PB2O-QM-8k: a smaller subset dataset of PB2O-QM containing 8776 drug/inhibitors containing C, H, O, N, F, Cl and/or S elements.
[b]ONIOM(MLP:ANI-2x) method was used for the molecules containing F, Cl and/or S elements, when AIQM1 was used.
[c]ONIOM(MLP:SE) method was used for the molecules containing B, P, Se, Br and/or I elements, when AIQM1 or ANI-2x was used.
[d]Number of the drugs/inhibitors.
Median absolute deviation (MAD) of the optimized bond distances, angles, rotatable dihedrals in the gas phase using AIQM1, ANI-2x, and GFN2-xTB (xTB) methods compared to the density functional theory (DFT, ωB97X-D/6-31 G(d)) method for the QR50, PB2O-QM-3 k, and PB2O-QM-8 k datasets.

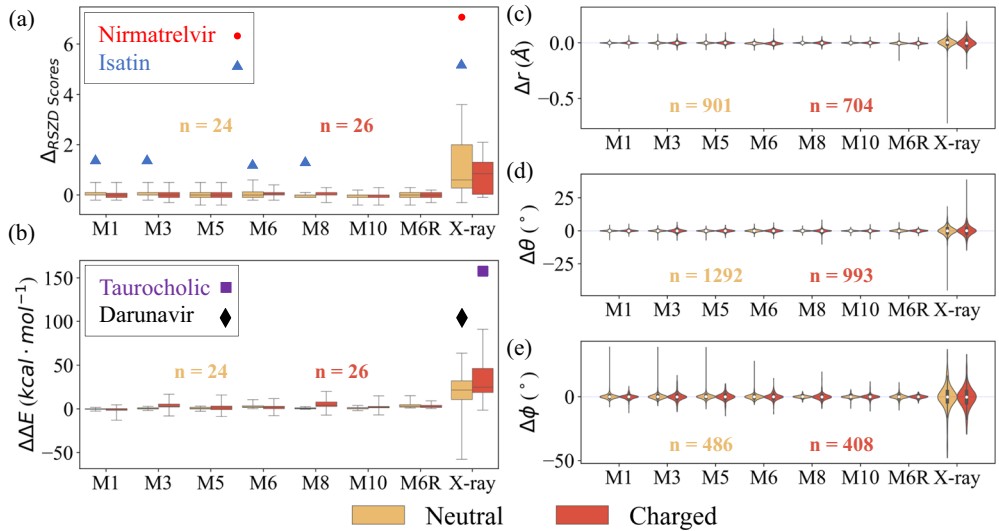

**Fig. 3 | Comparison of the quantum refinement results.** Boxplots of deviation of (**a**) real-space Z-difference (RSZD) scores as well as (**b**) strain energy (ΔΔE, kcal·mol⁻¹), in which the boxplots indicate median (center line), upper and lower quartiles (box limits), as well as whiskers (1.8× (X-ray) and 4.0× (QRs) × interquartile range in (**a**); 3.3× (X-ray) and 8.2× (QRs) interquartile range in (**b**)). Outlier points (with much larger deviations) were also marked using red circle, blue triangles, black rhombus, and purple square for four drugs/inhibitors. Most of these outliers are mainly attributed to their poor geometry in the X-ray crystal structure, whereas QRs on the Isatin system were found to have modestly larger deviations in four QR schemes. Violin plots of the deviations of (**c**) bond distances (Δr), (**d**) angles (Δθ) and (**e**) rotatable dihedrals (Δϕ) of drugs/inhibitors in the selected 50 protein–drug/inhibitor systems after **M1**–**M10** quantum refinement approaches compared to **M7** (ONIOM3(DFT:SE:MM)). The X-ray results were taken from the experimental structures without further refinement. The white dots in violin plots indicate the median values and the inner boxplots indicate the interquartile range. The number of data was given in each plot. Source data are provided as a Source Data file.

## Quantum refinement of protein–drug/inhibitor systems

These MLPs were further introduced into ONIOM-based QR schemes to refine the structures of the 50 protein–drug/inhibitor systems (Fig. 1 and Supplementary Figs. 1–4; the refined geometries are provided in Supplementary Data). Compared to the X-ray crystal structures, the real-space difference density Z (RSZD) scores of these 50 drugs/inhibitors (except nirmatrelvir, acylated ceftazidime, and isatin, including its covalent linkages) after various QRs were reduced by 1.0–1.1 on average (Fig. 3a and Supplementary Table 25), showing structural improvement by the QRs. The most

significant improvement was found in nirmatrelvir for one SARS-CoV-2 M^pro system and isatin for DJ-1 system, whose RSZD scores can be significantly decreased from 7.6/6.0 (X-ray crystal structure) to 0.5–1.0/0.9–2.1, respectively, due to the consideration of two conformers (*vide infra*).

In agreement with the above-mentioned smaller RSZD scores after QRs, the electron density analysis also demonstrated noticeable improvement of electron density around all refined drug/inhibitor binding sites generally (Fig. 4 and their corresponding electron density maps given in Supplementary Information). For instance, the electron

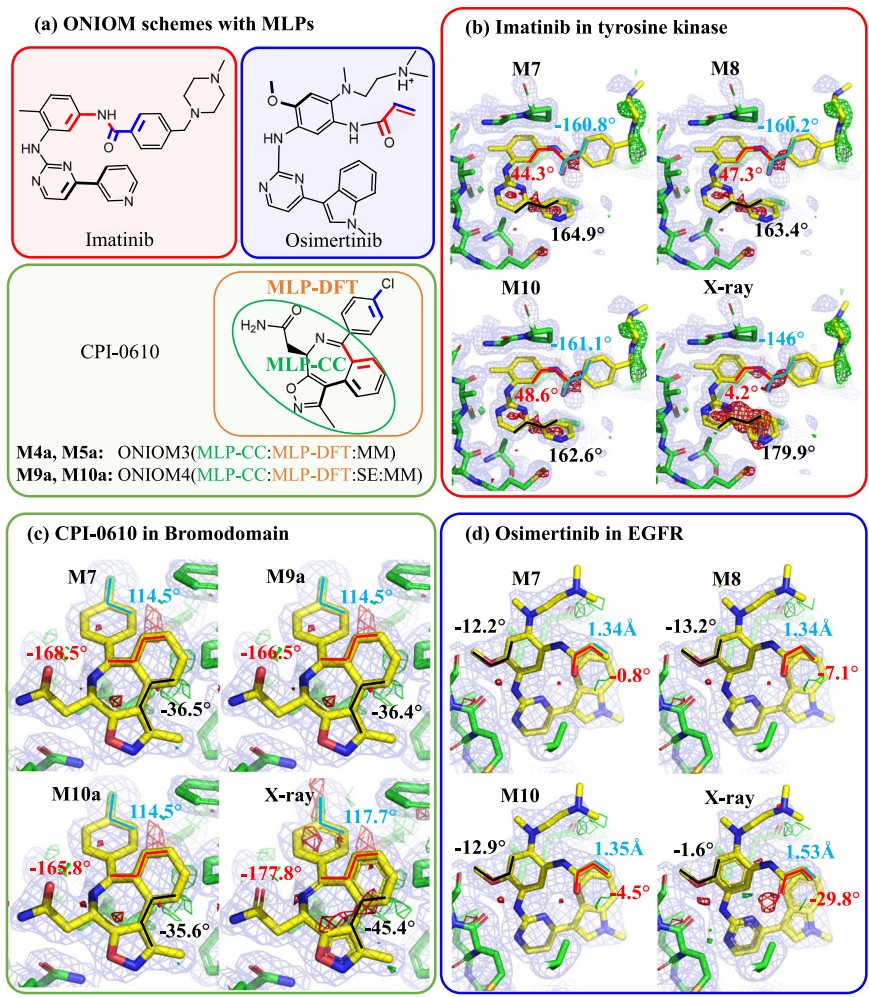

**Fig. 4 | Our own N-layered Integrated molecular Orbital and molecular Mechanics (ONIOM) scheme with machine learning potentials (MLPs), and electron density maps. a** Chemical structures of imatinib, CPI-0610 (with the ONIOM(MLP-CC:MLP-DFT) scheme) and osimertinib. Structures for the binding sites of (**b**) imatinib in spleen tyrosine kinase (PDB ID: 1XBB), (**c**) CPI-0610 in bromodomain (PDB ID: 5HLS) and (**d**) osimertinib in EGFR (PDB ID: 6JX4) from various quantum refinements using **M7, M8** (**M9a**), **M10** (**M10a**) and the X-ray crystal structures, including the electron density maps (2mF$_o$-DF$_c$ maps, contoured at 1.0 σ (blue), mF$_o$-DF$_c$ maps, contoured at +3.0 σ (green), and mF$_o$-DF$_c$ maps, contoured at −3.0 σ (red)). The X-ray results were taken from the experimental structures without further refinement. MLP-CC (coupled-cluster quality of MLP); MLP-DFT (density functional theory quality of MLP).

density maps for imatinib, osimertinib, and CPI-0610 systems showed better density fitting after our QRs. Moreover, compared to the X-ray electron density maps, the discrepancy with the experimental observations was significantly reduced (i.e., fewer green and/or red contours) by our QRs. In addition, for the case of the imatinib-tyrosine kinase system, the electron density maps around the imatinib binding site (Fig. 4b) are almost identical for refinements using the schemes based on DFT (ONIOM3(DFT:SE:MM), **M7**) and MLPs (ONIOM3(ANI-2x:SE:MM) and ONIOM3(AIQM1:SE:MM), **M8** and **M10**, respectively). This improved electron density surrounding the pyridine, pyrimidine and N-phenylbenzamide moieties may result from their changed dihedrals (black: from 179.9° (X-ray crystal structure) to 162.6°-164.9°; red: from 4.2° (X-ray crystal structure) to 44.3°-48.6°). Consequently, the negative RSZD score decreased from -1.8 (X-ray crystal structure) to −0.1-0.0 after various QRs (Supplementary Fig. 19).

In addition, the computed strain energies (Fig. 3b) of these 50 drugs/inhibitors were considerably reduced by 27.1-31.4 kcal·mol⁻¹ after various QRs on average. The greatest reduction in strain energy was observed in the taurocholic acid-CmeR (from 239.8 to approximately 78.7–85.6 kcal·mol⁻¹) and darunavir-HIV-1 (from 117.3 to approximately 14.1–20.4 kcal·mol⁻¹) systems after our QRs. This strain can be attributed to the underestimated C−S bond (b5: 0.18-0.22 Å shorter) of taurocholic acid and two C−C bonds (b₃ and b₄: 0.25-0.28 Å shorter) of darunavir in the X-ray structures (Fig. 1b, Supplementary Fig. 78, and Supplementary Table 18). Therefore, these computational results illustrate that all QR schemes (**M1–M10**) clearly improved the local drug/inhibitor binding sites compared to those in the X-ray crystal structures.

Moreover, our structural analysis (Fig. 3c−e) further revealed that the bond distances, angles, and rotatable dihedrals of the drugs/inhibitors after refinement by various QR schemes remained consistent with the most reliable ONIOM3(DFT:SE:MM) method (**M7**). Compared to the X-ray structures, darunavir, oseltamivir, and osimertinib structures refined using methods **M1–M10** displayed the largest absolute bond deviation (b₁: 0.19-0.21 Å, b₂: 0.18-0.20 Å, b₃: 0.28-0.29 Å, b₄: 0.25-0.26 Å; Supplementary Table 18) among the other drug/inhibitor systems, which can account for the very high strain energies computed in these X-ray structures (96.3–117.3 kcal·mol⁻¹, see Supplementary Table 26). Additionally, the largest absolute angle and dihedral deviations (using methods **M1–M10**) were found in our refined nirmatrelvir (a₁: 14.9°-17.5°) and imatinib (d₁: 37.7°-49.9°) structures (Supplementary Table 18), respectively.

## Combination of two MLPs through ONIOM

Since CC-level MLPs (ANI-1ccx and AIQM1) can only be applied to limited elements (H, C, N, O), these higher-level MLPs cannot be employed to describe broader drug/inhibitor systems with more elements (F, S, Cl). To overcome this limitation and describe more drug/inhibitor systems, two different levels of MLPs were combined for the first time through an extrapolative ONIOM scheme, in which the major core structures are described by the higher-level MLP-CC method (ANI-1ccx or AIQM1) and the remaining parts containing other elements are described by the lower-level MLP-DFT (ANI-2x) method (Fig. 4a). This new combination of two MLPs enables the unprecedented ONIOM3(MLP-CC:MLP-DFT:MM) (**M4a** and **M5a**) and ONIOM4(MLP-CC:MLP-DFT:SE:MM) (**M9a** and **M10a**) schemes for our QRs on the 20 selected protein–drug/inhibitor systems containing F, Cl or S element (Supplementary Fig. 3).

Pleasingly, our refined CPI-0610 structure in bromodomain obtained by these ONIOM4-based schemes can give similar electron density around the binding site to the reliable ONIOM3(DFT:SE:MM) **M7** scheme, with reduced discrepancy (i.e., fewer red and/or green contours) from the experimental observations (Fig. 4c and Supplementary Fig. 66). This improved electron density in the CPI-0610 system after these ONIOM4(ANI-1ccx:ANI-2x:SE:MM)- and ONIOM4(AIQM1:ANI-2x:SE:MM)-based (**M9a** and **M10a**, respectively) QRs might result from the changed dihedrals (black: from −45.4° (X-ray crystal structure) to −36.5°–−35.6°; red: from −177.8° to −168.5°–−165.8°) and changed $p$-ClC$_6$H$_4$ angle (blue: from 117.7° (X-ray crystal structure) to 114.4°-114.5°). Likewise, their RSZD scores for the CPI-0610 system were reduced from 1.3 (X-ray crystal structure) to 0.7–0.8 after these ONIOM4-based QRs. Moreover, the RMSD for the CPI-0610 structure refined by the ONIOM4(MLP-CC:MLP-DFT:SE:MM) scheme with reference to ONIOM3(DFT:SE:MM) (**M7**) was very small (bonds < 0.009 Å, angles < 0.8°, dihedrals < 2.6°). Similarly satisfactory refined results were also found in the other 19 systems (e.g., RSZD scores reduced by 0.4–7.1, Supplementary Table 25). Consequently, these findings show that these unique ONIOM4(MLP-CC:MLP-DFT:SE:MM) and ONIOM3(MLP-CC:MLP-DFT:MM) schemes can improve protein–drug/inhibitor structures (comparable to the DFT method) with higher computational efficiency, suggesting that the combination of several levels of MLPs via the ONIOM approach offers a promising avenue for overcoming some limitations of MLPs and enhancing their advantages.

## Combined MLP-CC/SE methods for broader drug molecules

Moreover, the higher-level AIQM1 and GFN2-xTB methods were further combined through the ONIOM scheme for our QRs on the three selected protein–drug/inhibitor systems (Supplementary Fig. 4), in which the major core of the drug/inhibitor structures and their remaining parts containing other elements (P or Br) are described by the AIQM1 and SE methods, respectively. The RMSD and MAD for these refined drug/inhibitor structures with reference to ONIOM3(DFT:SE:MM) (**M7**) were very small (RMSD: bonds < 0.011 Å, angles < 0.7°, dihedrals< 2.6°; MAD: bonds < 0.011 Å, angles < 0.5°, dihedrals < 2.3°). Moreover, our refined drug/inhibitor structures can give similar electron density around the binding site to the reliable **M7** scheme (Supplementary Figs. 186, 191, and 229). What's more, their RSZD scores were reduced from 1.4–3.9 (X-ray crystal structure) to 1.3–2.5 after these QRs. Therefore, these results show that the combined MLP-CC with SE method can further resolve the limitations of MLPs and be applied to much more elements and molecules.

## Correlation

Pleasingly, the use of these MLPs as the QM method successfully accelerated our QR processes with comparable accuracy to QM (e.g., RSZD score, strain energy and geometrical parameters). Additionally, MLP-CC based schemes, (**M4-M5** and **M9-M10**) gave the lowest RSZD scores among our ONIOM2- and ONIOM3-based QRs for 39 of the 50 selected systems (Supplementary Table 25), even lower than the DFT-based ONIOM3(DFT:SE:MM) **M7** scheme. Figure 5 displays the correlation ($R^2$) of the refined bond distances, angles, rotatable dihedrals, RSZD scores and strain energy obtained by the **M8, M10,** and **M6R** schemes (ONIOM3(ANI-2x:SE:MM), ONIOM3(AIQM1:SE:MM) and ONIOM2(SE:MM), respectively) compared to those obtained by the **M7** scheme. The refined bond distances and angles obtained by the **M8, M10** and **M6R** were in very good agreement with those obtained by the **M7** for the drug/inhibitor systems containing both charged and neutral groups (bond distances: $R^2 > 0.988$; angles: $R^2 > 0.962$). In contrast, their rotatable dihedrals, RSZD scores, and strain energies showed a clear disparity between the systems containing neutral and charged groups.

For neutral systems, MLP-based (ONIOM3(ANI-2x:SE:MM), ONIOM3(AIQM1:SE:MM), **M8 and M10**, respectively) and SE-based (ONIOM2-(SE:MM), **M6R**) schemes gave RSZD scores ($R^2 > 0.971$) consistent with those of the DFT-based scheme (ONIOM3(DFT:SE:MM), **M7**). Likewise, the **M8, M10** and **M6R** schemes also showed good agreement for the dihedrals ($R^2 > 0.935$). Moreover, the MLP-DFT-based (**M8**) and MLP-CC-based (**M10**) schemes gave similar strain energies to those of the DFT-based scheme ($R^2$: 0.996 and 0.992, respectively), significantly better than the those of SE-based scheme (**M6R**, $R^2$: 0.910). On the other hand, for the systems containing charged group(s), as ANI-type MLPs were not developed for charged

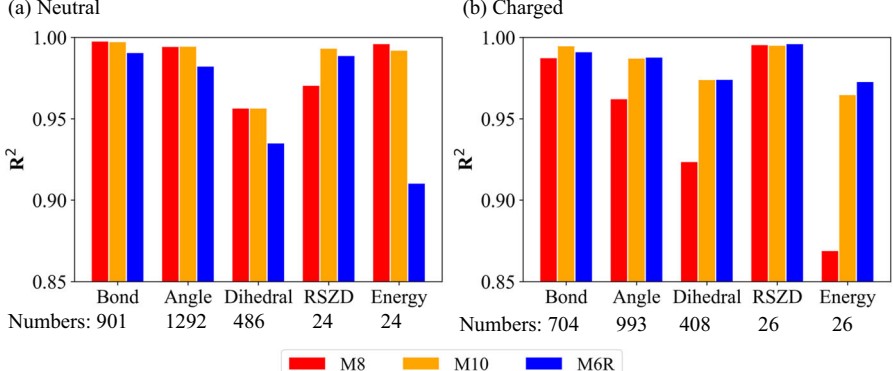

**Fig. 5 | Accuracy of different QR schemes.** Correlation ($R^2$) of the refined bond distances, angles, rotatable dihedrals, real-space Z-difference (RSZD) scores, and strain energy for the drugs/inhibitors containing (**a**) neutral and (**b**) charged group(s) by the **M8**, **M10** and **M6R** schemes (ONIOM3(ANI-2x:SE:MM), ONIOM3(AIQM1:SE:MM) and ONIOM2(SE:MM), respectively) compared to those obtained by the **M7** scheme (ONIOM3(DFT:SE:MM)). Numbers of data were given in each plot. Source data are provided as a Source Data file.

systems, the **M8** scheme gave the lowest $R^2$ values for the angles (0.962), dihedrals (0.924) and strain energy (0.869). However, all MLP-based schemes (**M8-M10**) still showed RSZD scores consistent with those of the DFT-based scheme (**M7**; $R^2$: 0.995–0.996), as well as the SE-based scheme (**M6R**, $R^2$: 0.996). Promisingly, the AIQM1-based scheme (**M10**) showed high $R^2$ on the bonds (0.995) and dihedrals (0.974). Therefore, compared to the reliable DFT-based scheme, these MLP-based schemes displayed good performance (structures, RSZD scores and strain energy) in our QRs of neutral systems, and the AIQM1-based scheme can offer high accuracy for the challenging charged cases.

### Structural comparison in the gas phase and in proteins
Figure 6 shows RMSDs (with reference to the DFT method) of all computed drug/inhibitor structures in the gas phase and in proteins. The RMSDs of the optimized drug/inhibitor structures in the gas phase (0.40–0.74 Å for ANI-2x, AIQM1, and GFN2-xTB) were significantly higher than their corresponding structures in proteins refined by QRs (0.02–0.03 Å: **M8**, **M10**, and **M6R** relative to **M7**). These results can be attributed to the much higher structural flexibility in the gas phase, compared to the confined space of the protein-binding sites. Moreover, the RMSDs for the drugs/inhibitors containing charged group(s) were found to be slightly larger than those for the neutral drugs/inhibitors when ANI-2x were used, possibly due to the aforementioned limitation of ANI methods. Encouragingly, the higher-level AIQM1 method can give better performance of the optimized drug/inhibitor structures in the gas phase (charged: 0.49 Å; neutral: 0.59 Å) than the ANI-2x method (charged: 0.74 Å; neutral: 0.67 Å). Overall, these results demonstrate that QRs using MLPs (especially the more accurate CC-level AIQM1) can provide reliable drug/inhibitor structures in proteins (comparable to the DFT level), although larger structural errors were observed in the gas phase.

### Efficiency
In terms of computational costs, QRs using these MLPs are significantly more efficient than the DFT method as the high level (Table 3). For instance, QRs of imatinib in spleen tyrosine kinase required approximately 144 CPU core-hours (Intel Xeon Gold 5218) for the DFT-based (ONIOM2(DFT:MM), **M1**) scheme, but only about 1.5–2.4 core-hours for the MLP-based (ONIOM2(ANI-2x:MM) and ONIOM2(AIQM1:MM), **M3** and **M5**, respectively) and SE-based (ONIOM2(SE:MM), **M6**) schemes. Alternatively, QR using the robust **M3** scheme was first performed, followed by additional refinement using **M1** (**M3→M1**). Approximately 54.8 CPU core-hours were required in this dual refinement approach to reach the genius DFT accuracy. For the case of nirmatrelvir in the SARS-CoV-2 M$^{Pro}$, QRs using the MLP-based schemes (**M3**, **M5** and **M6**) required about 6.7–31.5 core-hours, which was much faster than using the DFT-based **M1** scheme (3948.4 core-hours). A

dual refinement approach (**M3→M1**) also significantly reduced the QR time (626.8 core-hours). These benchmark results demonstrate that QRs using MLPs can give reliable results with much lower computational costs. Computational time can also be reduced using a dual refinement approach using MLPs followed by the DFT method as the high level.

### Two conformers of nirmatrelvir in SARS-CoV-2 M$^{Pro}$
The electron density map around the nirmatrelvir binding site in the crystal structure of wild-type SARS-CoV-2 M$^{Pro}$ shows pronounced red contours around the C–S covalent bond linkage to the Cys145 (Fig. 7a). Unfortunately, our QRs performed by several schemes (**M1–M10**) only marginally alleviated this substantial discrepancy in the electron density and slightly reduced the RSZD score from 7.6 (X-ray structure) to 5.8 (Supplementary Table 38). The electron density discrepancy in the C–S bond linkage after these QRs remained severe (Supplementary Fig. 42). On the basis of the reversible C–S bond formation mechanism[52], the possible existence of both bonded and nonbonded conformers (Fig. 7a, b) was further considered in our QRs.

These two conformers were first individually refined using the DFT-based ONIOM3(DFT:SE:MM) **M7** scheme. The refined results were then combined with different ratios of occupation of the two conformers. Our refined results suggest that an occupation ratio of approximately 7:3 (bonded: nonbonded) gives the greatest structural improvement with the lowest RSZD scores (0.5, Fig. 7b) as well as the marked improvement of the electron density (Fig. 7c). Similarly, ONIOM QRs using the other MLP-based schemes (ONIOM3(ANI-2x:SE:MM), ONIOM3(ANI-1ccx:SE:MM) and ONIOM3(AIQM1:SE:MM), **M8-M10**, respectively) and the same occupation ratio (7:3) also provided significant improvement in the electron density (Fig. 7c) and the

**Table 3 | Efficiency of the quantum refinements**

| System | M1 | M3 | M5 | M6 | M3→M1 |
|---|---|---|---|---|---|
| Imatinib | 144.0 | 1.9 | 2.4 | 1.5 | 54.8 |
| Rivaroxaban | 188.4 | 53.1 | 50.9 | 15.0 | 186.1 |
| Oseltamivir | 1004.0 | 56.9 | 60.8 | 47.7 | 232.6 |
| Ciprofloxacin | 5987.6 | 350.8 | 426.9 | 323.2 | 2994.1 |
| Mometasone | 1328.1 | 162.2 | 1061.6 | 153.9 | 1188.1 |
| CPI-0610 | 168.8 | 3.2 | 204.7 | 23.1 | 129.4 |
| Darunavir | 185.8 | 158.5 | 67.1 | 73.0 | 117.6 |
| Ibrutinib | 541.4 | 171.8 | 401.0 | 46.8 | 253.0 |
| Osimertinib | 5798.5 | 812.3 | 265.5 | 198.7 | 196.3 |
| Nirmatrelvir | 3948.4 | 6.7 | 10.3 | 31.5 | 626.8 |

Computational cost (CPU core-hours) of different quantum refinements for the ten selected systems shown in Fig. 1. All calculations are performed on an Intel Xeon Gold 5218 CPU.

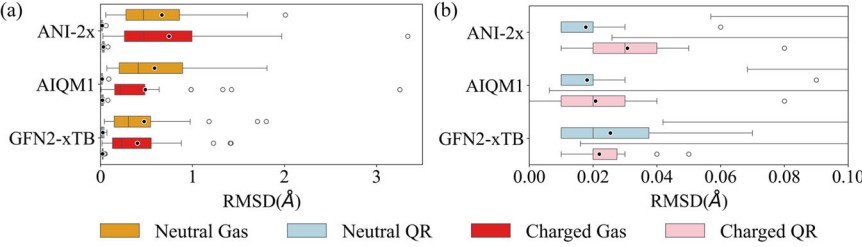

**Fig. 6 | Structural comparison in the gas phase and in proteins.** Boxplots of root mean square deviation (RMSD) of structures optimized using the ANI-2x, AIQM1, and (SE) GFN2-xTB methods compared to the (DFT) ωB97X-D method for the neutral drugs/inhibitors (26 structures) in the gas phase (orange) and in the proteins after quantum refinement (blue) and for the charged drugs/inhibitors (26 structures) in the gas phase (red) and in the proteins after quantum refinements (pink). **a** Those with RMSD range from 0 to 3.5 Å; (**b**) Those with RMSD range from 0 to 0.1 Å. The boxplots indicate median values, interquartile range, minimum and maximum value, and individual data points. The black dots indicate the average values. The QR data by the **M7, M8, M10** and **M6R** schemes (ONIOM3(DFT:SE:MM), ONIOM3(ANI-2x:SE:MM), ONIOM3(AIQM1:SE:MM) and ONIOM2(SE:MM), respectively) were used. Source data are provided as a Source Data file.

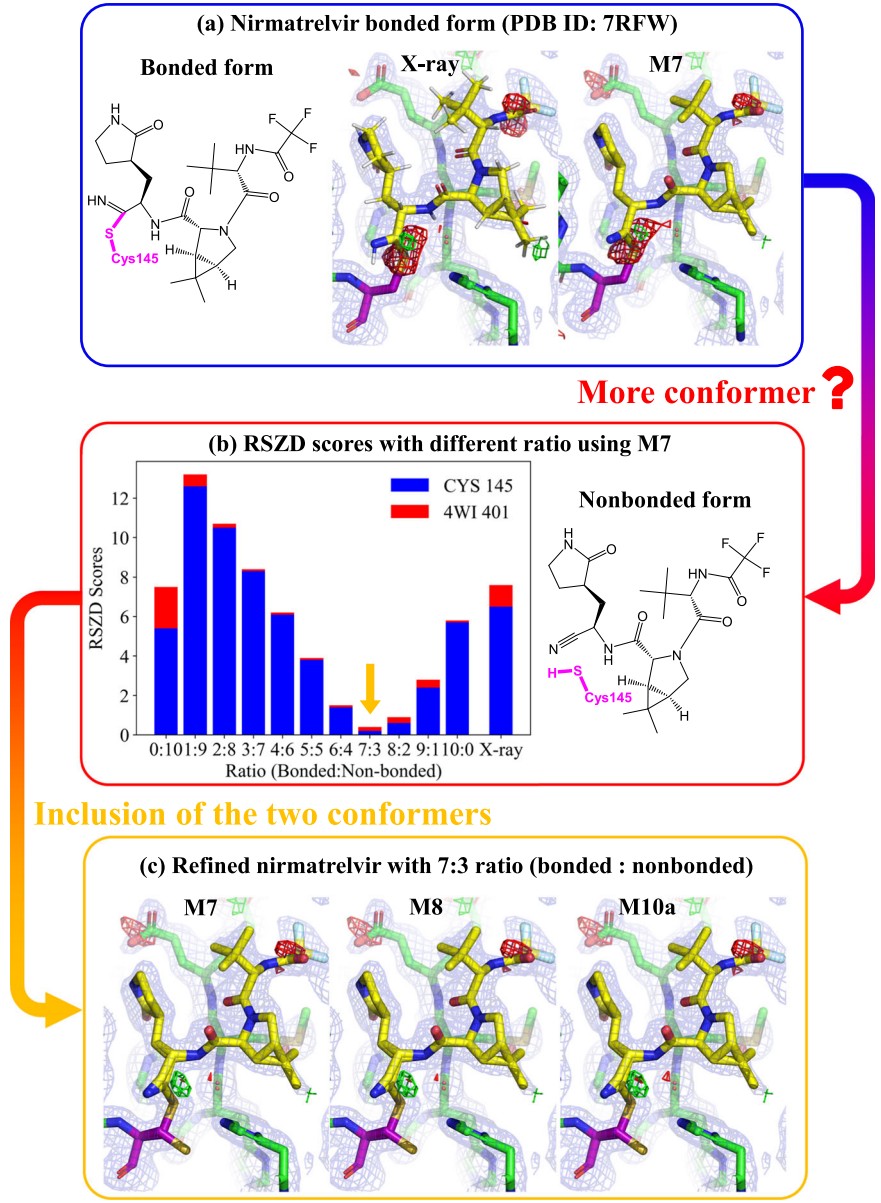

**Fig. 7 | Quantum refinements of SARS-CoV-2 M^Pro. a** Nirmatrelvir bonded to Cys145 in the X-ray structure and refined by **M7** quantum refinement scheme, including the electron density maps. **b** Nirmatrelvir nonbonded to Cys145 and real-space Z-difference (RSZD) scores of the Cys145 residue and nirmatrelvir (4WI) in wild-type SARS-CoV-2 M^Pro refined by **M7**-based quantum refinement with different occupations (bonded: nonbonded). **c** Structures of the nirmatrelvir binding site in SARS-CoV-2 M^Pro refined by various quantum refinement schemes (**M7, M8**, and **M10a**) with an occupation ratio of 7:3 (bonded: nonbonded), including the electron density maps (2mF_o-DF_c maps, contoured at 1.0 σ (blue), mF_o-DF_c maps, contoured at + 3.0 σ (green), and mF_o-DF_c maps, contoured at −3.0 σ (red)). The X-ray results were taken from the experimental structure without further refinement. Source data of panel (**b**) are provided as a Source Data file.

RSZD scores (from 7.6 (X-ray crystal structure) to 0.5−0.7 only). Moreover, the computed strain energy was substantially decreased by 30.2−36.8 kcal·mol⁻¹. Again, the refined results using the MLP-based schemes (**M8**-**M10**) were very similar to those using the DFT-based scheme (**M7**) with derivation of RSZD scores < 0.2 only and almost identical electron density maps (Fig. 7c). Consequently, our QR results provide computational evidence of the coexistence of the bonded and nonbonded forms of nirmatrelvir in the crystal structure of SARS-CoV-2 M^Pro, which should afford important structural information for designing better SARS-CoV-2 drugs. Moreover, large RSZD score and electron density discrepancy of the key C−S bond linkage were observed in another crystal structure of wild-type SARS-CoV-2 M^Pro (PDB ID: 7SI9, Supplementary Table 42 and Supplementary Fig. 52), which may also imply the coexistence of the bonded and nonbonded forms. Furthermore, the coexistence of the bonded and nonbonded

forms were recently proposed in another crystal structure of wild-type SARS-CoV-2 M^Pro (PDB ID: 7VH8)[52].

## Discussion

In this study, machine learning potentials (MLPs) were proposed and utilized to replace the reliable but expensive QM method and to accelerate multiscale QR processes for the first time. To overcome the element restrictions in some MLPs, two different levels of MLPs were also combined by an extrapolative ONIOM approach and then applied as the unprecedented ONIOM3(MLP1:MLP2:MM) and ONIOM4(MLP1:MLP2:SE:MM) schemes for QR. Our QR results demonstrated that MLPs (especially the high-level AIQM1 method reaching coupled-cluster accuracy) could achieve highly accurate drug/inhibitor structures in 50 different protein−drug/inhibitor systems, comparable to the results of the reliable DFT method, with

computational costs significantly lower by roughly two orders of magnitude. Moreover, our QR results provided computational evidence of coexisting the bonded and nonbonded forms of the FDA-approved drug nirmatrelvir in one crystal structure of SARS-CoV-2 M[Pro], which should provide new structural insights for drug design for SARS-CoV-2 treatment. Our proof-of-concept study showed that powerful MLPs can accelerate the QR of protein–drug/inhibitor complexes with high accuracy, which should promote more QR applications and provide new atomistic insights into molecular recognition, catalysis, and drug development. Additionally, apart from X-ray crystallography, we believe that MLPs could be helpful for modern structural determination methods of biomacromolecules (e.g. Cryo-EM, MicroED)[53,54]. Furthermore, computational benchmark datasets (PB20-QM, PB20-QM-8 k and PB20-QM-3 k) were set up to evaluate the structural reliability of 3–19 k drug/inhibitor molecules computed by the DFT, MLPs and/or SE methods, which hopefully help future development of better DFT, MLPs and/or SE methods for drug/inhibitor molecules. We are also optimistic that with the advances of more high-level and general MLPs developed by different groups[55–59], the fast and reliable QR of diverse biosystems will be routinely performed on any normal desktop computer in the future.

## Methods

### Multiscale QR methods using MLPs

On the basis of our previous ONIOM-based QR method[22], the total energy function of the entire system is given by Eq. 1:

$$E_{total} = E_{ONIOM} + \omega_\alpha * E_{xray} \tag{1}$$

$$E_{ONIOM2(high:low)} = E_{high,model} + E_{low,real} - E_{low,model} \tag{2}$$

$$E_{ONIOM3(high:medium:low)} = E_{high,model} + E_{medium,intermediate} - E_{medium,model} + E_{low,real} - E_{low,intermediate} \tag{3}$$

where $E_{ONIOM}$ represents the energy contribution from ONIOM-based calculations[60], $E_{xray}$ stands for the energy contribution derived from the crystallographic penalty and $\omega_\alpha$ is the weighting factor that balances the contributions of each term. Equations 2 and 3 represent the two- and three-layer ONIOM schemes, respectively.

In general, flexible ONIOM methods can significantly improve the structure of active sites by ONIOM2(QM:MM), ONIOM2(SE:MM), and/or ONIOM3(QM:SE:MM) schemes. To reduce the computational cost of the QM method and to accelerate the ONIOM-based QRs, a few MLPs (ANI-2x, ANI-1ccx, and AIQM1)[33,35,36] are employed to replace the QM method, i.e., ONIOM2(MLP:MM) and ONIOM3(MLP:SE:MM) schemes. However, CC-level MLPs (MLP-CC, such as ANI-1ccx and AIQM1) were only trained on systems containing H, C, N, and O elements, while DFT-level MLPs (MLP-DFT, such as ANI-2x) was trained on systems containing H, C, N, O, F, S, and Cl elements. To extend the MLP-CC applications to more drug/inhibitor molecules containing F, S, Cl elements, two different levels of MLPs were further combined through an extrapolative ONIOM approach (ONIOM2(MLP-CC:MLP-DFT), Eq. 4). Such ONIOM2(MLP-CC:MLP-DFT) part was then used to replace the $E_{high,model}$ part in ONIOM2(QM:MM) in Eq. 2 or that in ONIOM3(QM:SE:MM) in Eq. 3 to derive unique ONIOM3(MLP-CC:MLP-DFT:MM) and ONIOM4(MLP-CC:MLP-DFT:SE:MM) schemes, respectively. To further apply MLP to even boarder drug/inhibitor molecules containing P or Br element, AIQM1 (or ANI-2x) and SE methods were also combined to describe the drug/inhibitor molecules by the ONIOM approach (Eq. 5), which replaces the $E_{high,model}$ part in

ONIOM2(QM:MM) in Eq. 2 or that in ONIOM3(QM:SE:MM) in Eq. 3.

$$E_{ONIOM2(MLP-CC:MLP-DFT)} = E_{MLP-CC,model} + E_{MLP-DFT,real} - E_{MLP-DFT,model} \tag{4}$$

$$E_{ONIOM2(MLP:xTB)} = E_{MLP,model} + E_{xTB,real} - E_{xTB,model} \tag{5}$$

Combination of two different MLP levels can readily be applied and extended to an additive QM/MM framework (Eqs. 6–7)[60–62].

$$E_{MLP-CC/MLP-DFT/MM} = E_{MLP-CC,model} + E_{MLP-DFT,intermediate} - E_{MLP-DFT,model} + E_{MM} + E_{MLP-DFT-MM} \tag{6}$$

$$E_{MLP-CC/MLP-DFT/xTB/MM} = E_{MLP-CC,model} + E_{MLP-DFT,intermediate1} - E_{MLP-DFT,model} + E_{xTB,intermediate2} - E_{xTB,intermediate1} + E_{MM} + E_{xTB-MM} \tag{7}$$

The experimental part $E_{xray}$ (Eq. 1) was obtained from the Crystallography and NMR System (CNS) program[63].

### Protein preparations

All experimental data and protein–drug/inhibitor crystal structures were obtained from the Protein Data Bank (PDB)[64–72]. The CNS topology and parameter files of these drug/inhibitor molecules can be generated from PRODRG and ATB servers. Prior to ONIOM QR calculations, the proteins were first prepared by determining the protonated state and the rotamer of some amino-acid side chains (see details in the Supporting Information), as well as addition of hydrogen atoms. The rotamer of the amino-acid side chain was determined by WHATCHECK and our visual examination. The addition of hydrogen atoms and optimization of the hydrogen-bond network were then performed using PDB2PQR 3.5.2 program. The protonated states of the titratable residues were assigned on the basis of estimated pKa results computed from PROPKA 3.0 program at the pH of crystallization, while the protonated states of the drug/inhibitor molecules were generally assigned by cross-validation of MolGpka, Graph-pKa, and pkasolver. All added hydrogens were firstly optimized using ONIOM2(DFT:MM) with fixing the heavy atoms.

### ONIOM QR calculations

Various ONIOM-based schemes were used to assess performance of our QRs (**M1**–**M10**, Table 1), after the protein preparations. All QRs were conducted using our ONIOM_QR code[22]. Various ONIOM-based ONIOM2(QM:MM), ONIOM2(MLP:MM), ONIOM2(SE:MM), ONIOM3(QM:SE:MM) and ONIOM3(MLP:SE:MM) schemes were mainly performed using Gaussian 16. A popular ωB97X-D functionals as the DFT method combined with 6–31 G(d) basis sets was employed as the QM method[73]. ANI-1ccx[33], ANI-1x[34], ANI-2x[35], and AIQM1[36] methods were used as MLPs. GFN2-xTB method was employed as the SE method[74], and Amber ff14SB force fields served as the MM method[75]. The ANI and AIQM1 methods were performed using TorchANI[76] and MLatom[77], respectively, which were called through Gaussian external interface to achieve two-layer ONIOM2(MLP-CC:MLP-DFT) (such as ONIOM2(ANI-1ccx:ANI-2x) and ONIOM2(AIQM1:ANI-2x)), three-layer ONIOM3(MLP-CC:MLP-DFT:MM) and four-layer ONIOM4(MLP-CC:MLP-DFT:SE:MM) schemes. Additionally, optimization of these drug/inhibitor molecules in the gas phase using ANI-2x, ANI-1x, ANI-1ccx, AIQM1, GFN2-xTB, and ωB97X-D/6-31 G(d) as well as, for some molecules containing P, Br elements, ONIOM2(AIQM1:GFN2-xTB) methods were carried out to analyze the MLPs performance using Gaussian 16 as the geometry optimizer. Moreover, effects of dispersion correction, DFT functional, and diffuse functions in the QM method on our QRs of the selected 10 drug/inhibitor (shown in Fig. 1) were

examined. These selected 10 drug/inhibitor-protein structures refined based on **M1** quantum refinements using ωB97X/6-31 G(d), ωB97X-D/6-31 + G(d) or M06-2X/6-31 G(d) method as the QM method are very similar as those by ωB97X-D/6-31 G(d) method (MAD: 0.001 Å, 0.0–0.1°, and 0.2–0.3°, respectively; Supplementary Table 3).

For our two- and three-layer ONIOM QR calculations/setup (see the Supplementary Information for details), the drug/inhibitor molecules (except Nirmatrelvir in wild-type SARS-CoV-2 M$^{Pro}$, ceftazidime in beta-lactamase, and isatin in DJ-1) were set as the high-layer model part and the optimized region, as well as the medium layer in the three-layer ONIOM methods included neighboring residues within a radius of 3.0 Å to the model part. In the case of the SARS-CoV-2 M$^{pro}$ system, the Nirmatrelvir drug and its linkage CYS145 residue were defined as the high-layer model part and optimized region. Similarly, in the case of the beta-lactamase and DJ-1 systems, the drug/inhibitor and its covalent linkage residue were defined as the high-layer model part and optimized region. For the systems containing H, C, N, and O elements, **M4, M5, M9** and **M10** QR schemes were used. Whereas, for the 20 drug/inhibitor systems containing H, C, N, O, F, Cl, and S elements, **M4a, M5a, M9a,** and **M10a** QR schemes were used. ONIOM(MLP:SE) method was used for the three molecules containing P and Br elements when AIQM1 or ANI-2x was used. Furthermore, the highest-methods combination used in this study was considered to be the most reliable computational method: ONIOM3(DFT:SE:MM) (**M7**). All refined results by different methods were generally used to compare with those by **M7**.

The convergence conditions (atomic unit) for our QR calculations were kept consistent with the default settings in Gaussian 16: $\Delta E < 10^{-5}$, Step$_{max}$ < $1.8 \times 10^{-2}$, Step$_{RMS}$ < $1.2 \times 10^{-2}$, $\nabla E_{max}$ < $4.5 \times 10^{-3}$, $\nabla E_{RMS}$ < $3 \times 10^{-3}$. The weighting factor $\omega_\alpha$ was derived from CNS for each system (see details in the Supporting Information). Analysis on the refined results (including electron density, and real-space difference density Z (RSZD)) score were carried out using Refmac5, and Edstats modules implemented in CCP4i2 8.0. Electron density maps were drawn using Pymol. The strain energy was determined by calculating the QM energy difference between the fully-optimized ligand in the gas phase and the refined ligand extracted from the protein at the ωB97X-D/6-31 G(d) level.

## Gas-phase geometry optimization

All drug/inhibitor molecules were also optimized by the above-mentioned DFT, MLPs, and SE methods using Gaussian 16 as the geometry optimizer. In addition, effects of dispersion correction, DFT functional and diffuse functions were also examined using the selected 10 drug/inhibitor (shown in Fig. 1). These molecules optimized by ωB97X/6-31 G(d), ωB97X-D/6-31 + G(d) or M06-2X/6-31 G(d) method are very similar as those by ωB97X-D/6-31 G(d) method (MAD: 0.001 Å, 0.1–0.2°, and 7–16°, respectively; Supplementary Fig. 6, and Supplementary Table 2). Therefore, ωB97X-D/6-31 G(d) was used as the DFT method in this study.

## Datasets setup

Our drug/inhibitor molecules downloaded from PDBbind v2020 dataset were used to set up a benchmark dataset (denoted as PB20-QM), which contains 12,963 drug/inhibitor molecules after removing metal-containing systems. The protonation states of all these molecules remain preserved as those in the PDBbind v2020 dataset, except for minor corrections (to fix very poor or wrong positions of H atoms based on their corresponding PDB structures) in some molecules. Classification of charged group(s) in these molecules was based on protonation state(s) given in the PDBbind v2020 dataset. Geometry optimization for all molecules in this PB20-QM dataset in gas phase were conducted by the above-mentioned DFT and SE methods. In addition, two smaller sub-datasets (PB20-QM-8k: 8776 molecules containing, C, H, O, N, F, S, and/or Cl elements; and PB20-QM-3k: 3156

molecules mainly containing, C, H, O, and/or N elements) were also set up. Geometry optimization for all molecules in this PB20-QM-8k dataset were conducted by the above-mentioned DFT, SE, and ANI-2x methods, whereas those in PB20-QM-3k dataset were performed by the above-mentioned DFT, SE, AIQM1, and ANI-2x methods.

## Data availability

The crystal structures and experimental data used in this work are available in the RSCB Protein Data Bank (https://www.rcsb.org/) under the PDB IDs listed in the paper. All the supporting data are provided in the main text and Supplementary Information. Our PB20-QM benchmark datasets have been deposited in our GitHub repository (https://github.com/oscarchung-lab/PB20-QM). The optimized/refined geometries of the QR50 dataset in this study are provided in Supplementary Data file. Source data are also provided with this paper. Source data are provided with this paper.

## Code availability

All original code is publicly available at https://github.com/oscarchung-lab/ONIOM_QR/releases 2.0.0 as well as Zenodo (https://doi.org/10.5281/zenodo.10828284)[78].

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

## Acknowledgements

We gratefully acknowledge the financial support from the National Natural Science Foundation of China (21873043, 21933003, 22193020, and 22193023), Southern University of Science and Technology, the Shenzhen Nobel Prize Scientists Laboratory Project (C17783101), Guangdong Provincial Key Laboratory of Catalysis (2020B121201002), and Natural Science Foundation of Shenzhen Innovation Committee (JCYJ20220530115408019). We thank the Center for Computational Science and Engineering of Southern University of Science and Technology CHEM HPC at SUSTech for partly supporting this work. Mr. Hui Xia is acknowledged for his discussion during the early stage of this study.

## Author contributions

L.W.C. conceived and designed the project. Z.Y. carried out QR calculations. D.W. performed the optimizations in the gas phase. L.W.C. and X.L. supervised the project. Z.Y. and L.W.C. prepared the manuscript. All authors analyzed and discussed the results as well as assisted manuscript preparation.

## Competing interests

The authors declare no competing interests.
