## [Peer Review File · Nature Communications]

Accelerating Reliable Multiscale Quantum Refinement of Protein–Drug Systems Enabled by Machine LearningREVIEWER COMMENTS

Reviewer #1 (Remarks to the Author):

The manuscript "Accelerating Reliable Multiscale Quantum Refinement of Protein–Drug Systems Enabled by Machine Learning" describes the efforts of the authors to reduce the computational cost associated with the structural refinement step in crystallography. The main idea of the manuscript is to avoid expensive QM-based schemes in the refinement step by using machine learning potentials. The methodology described in the paper shows promising results. However, the following points need to be addressed before the paper can be published in Nature Communications.

1. The authors tested the validity of their methodology by using a benchmark dataset of 10 drug-protein complexes. This is a rather small dataset, insufficient to prove how useful the proposed methodology is. In order to show that the approach is a feasible alternative to established QM-based refinement methods, the methodology must be tested in a much larger set of compounds.
2. The chemical space covered by the test cases is an issue. Taking into account the vast chemical space covered by drug-protein complexes, the authors should explain how the current set of complexes (or in light of the previous point, the new set of benchmark complexes) covers at least the most frequent functional groups in drug discovery. In addition, the authors should also explain the reason for not including any phosphorus-containing drug in their study.
3. As reference for the geometry optimizations in the gas-phase, the authors chose the ω B97X functional. The reasoning that this functional was the one used to train the ANI-2x method is not convincing enough. The authors should soundly justify why this functional was selected as reference for the calculations. In addition, the used basis set was 6-31g(d), which lacks diffuse functions. Diffuse functions are often important to properly describe hydrogen bond interactions. The authors should check if changing the functional and basis set used for the optimizations affects the quality of the results in terms of agreement between the optimized geometries.
4. In line 294, the authors state that: "Moreover, the RMSDs for the charged drugs were found to be larger than those for the neutral drugs when MLPs were used...". This is, however, not the case for the AIQM1 MLP method (according to Figure 6), where the RMSD is larger for the neutral species compared to the charged ones. The authors should discuss this result.
5. The analysis of the efficiency of the method based on the CPU core-hours employed must be extended to all systems studied and not only to the two systems presented in Table 2.
6. The nomenclature used by the authors to refer to each of the methods in the text is difficult to follow if the reader is not constantly looking at Table 1. This issue could be ameliorated if the method is explicitly mentioned more often in the text. Even if this would slightly increase the length of the manuscript, it will definitely improve its readability.

The manuscript could be published in Nature Communications after major revisions, especially after the extension of the dataset mentioned above.

Reviewer #2 (Remarks to the Author):

Reviewer #3 (Remarks to the Author):

In this manuscript, Yan et al. report their computational development of multi-scale quantum refinement method. This method aims to improve or correct X-ray crystal structures of ten common drug molecules within the corresponding target protein structures by replacing robust machine learning potentials (MLPs; mainly ANI- or AIQM1-type) with more expensive yet reliable QM methods such as DFT or high-level CCSD(T) method, for the first time. Moreover, their combination of two different MLPs levels within the substrative ONIOM framework (i.e. ONIOM3(MLP1:MLP2:MM) and ONIOM4(MLP1:MLP2:SE:MM)) is novel and could inspire other computational chemists to adopt similar approach for other chemical and biochemical studies. Their quantum refinement results indicate that the proposed approach can give reliable and highly efficient results comparable to QM-level accuracy, especially the computationally prohibitive CCSD(T) method. A significant finding from this study is their computational evidence of the coexistence of bonded and nonbonded forms of the famous nirmatrelvir drug in the SARS-CoV-2 protein: revision/correction of this drug-protein structure. As SARS-CoV-2 is still a global issue and has attracted tremendous attention in different areas and communities (beyond just chemistry and biochemistry), this revised/corrected nirmatrelvir-SARS-CoV-2 protein structure should be of broad interest to readers of Nat. Commun. and other communities. Overall, this study likely advances biological structural refinement method and multi-scale simulation methods. I recommend the manuscript to be published in Nature Communications after a minor revision as follows,

- (1) It would be essential to give some more discussion on the limitations of MLPs. Is it possible to test one or two other MLPs besides the ANI and AIQM1 types?
- (2) The combination of two different MLP levels using the substrative ONIOM approach is interesting. Could such combination be applied to popular additive QM/MM method?
- (3) The ω B97X functional with the 6-31G(d) basis set was used in this study. It is more reliable to test the effect of the empirical dispersion corrections, using the ω B97X-D functional, at least for the drug structures in gas phase.
- (4) The authors discussed the computational costs in the ONIOM2 approach. Could the authors also compare the cost for the ONIOM3 method to provide a more comprehensive comparison?
- (5) SARS-CoV-2 is a virus name. The authors should give its protein name instead. Could the authors check any other available crystal structures of the same drug-protein that contain both the two bonded forms?
- (6) I recommend to cite some related recent QM/MM methods combined with MLPs or NLPs for drugs, such as J. Phys. Chem. Lett. 2018, 9, 3232; J. Chem. Inf. Model. 2021, 61, 1066; Phys. Chem. Chem. Phys. 2022, 24, 1326; Phys. Chem. Chem. Phys. 2022, 24, 18559.

Dear the reviewers,

We would like to sincerely thank the three reviewers for their effort to review our manuscript as well as their insightful and constructive comments, which significantly improves the quality of our work. We have made significant efforts for addressing all of the reviewers' comments by performing additional extensive calculations in our revised manuscript and SI, e.g.,

- 1) adding dispersion correction and/or diffuse functions to enhance reliability of the DFT method;
- 2) testing another reliable DFT method or another MLP;
- 3) constructing a new computational benchmark dataset for drug/inhibitor molecules (PB20-QM; its sub-datasets: PB20-QM-3k and PB20-QM-8k) from the PDBbind v2020 database and conducting extensive benchmark (gas-phase) calculations of 3,156-12,963 drug/inhibitor molecules by DFT, MLPs and xTB methods;
- 4) searching 40 more protein systems containing other drug/inhibitor molecules with 37 out of the 40 reported most common functional groups followed by performing multiscale quantum refinements;
- 5) demonstrating and discussing the limitations of MLPs.

As some of our new protein systems contain inhibitors, we have changed to use “drug/inhibitor” or “drugs/inhibitors” in this revised work, instead of “drug” or “drugs”. Moreover, we have updated all results and analysis based on our latest calculations using ω B97X-D as the DFT method. We list out our point-by-point response and detailed action as follows. We hope that you find our new calculations/comparison and revision acceptable or satisfactory.

Response to Reviewer #1

Comments: The manuscript "Accelerating Reliable Multiscale Quantum Refinement of Protein–Drug Systems Enabled by Machine Learning" describes the efforts of the authors to reduce the computational cost associated with the structural refinement step

in crystallography. The main idea of the manuscript is to avoid expensive QM-based schemes in the refinement step by using machine learning potentials. The methodology described in the paper shows promising results. However, the following points need to be addressed before the paper can be published in Nature Communications.

Our Response: Thank you so much for your positive comments and valuable suggestions. We have tried our best to conduct much more calculations and analysis to address your concerns.

Comment 1: The authors tested the validity of their methodology by using a benchmark dataset of 10 drug-protein complexes. This is a rather small dataset, insufficient to prove how useful the proposed methodology is. In order to show that the approach is a feasible alternative to established QM-based refinement methods, the methodology must be tested in a much larger set of compounds.

Our Response: Thank you so much for your important comment. We agree with you that our dataset of 10-systems is small. Notably, based on our results, the reliability of MLPs compared to the reliable QM method is the most critical factor in the accuracy of our multiscale quantum refinements. To address your concerns and streamline our extensive calculations, we have set up a new computational benchmark dataset for drug/inhibitor molecules (PB20-QM; its sub-datasets: PB20-QM-3k and PB20-QM-8k) downloaded from the PDBbind v2020 database and conducted extensive benchmark (gas-phase) calculations of 3,156-12,963 drug/inhibitor molecules by DFT, MLPs and xTB methods to compare the geometrical accuracy. 12,963 drug/inhibitor molecules in PB20-QM dataset were optimized by the ω B97X-D and xTB methods. Also, 8776 molecules containing, C, H, O, N, F, S, and/or Cl elements only in PB20-QM-8k dataset were optimized by DFT-type ANI-2x method, and 3156 molecules mainly containing, C, H, O, and/or N elements only in PB20-QM-3k dataset were optimized by coupled-cluster-type AIQM1 method, due to element limitation of MLPs. For some cases containing heavier elements, ONIOM(AIQM1:xTB) method was also applied to overcome the element limitation of AIQM1. Our new benchmark calculations again show that these MLPs and ONIOM(AIQM1:xTB) methods can give reliable structures.

In addition, we have uploaded our new computational dataset for drug/inhibitor molecules (PB20-QM) to our github website (<https://github.com/oscarchung-lab/PB20-QM>), which should be useful for future development of better force fields, MLPs and/or DFT methods for drug/inhibitor molecules by different groups. Moreover, we have extended our quantum refinements to 50 drug-protein complexes with much more different functional groups (see our below response 2). We have added these new results and discussion in our revised manuscript and SI as follow:

Our Action:

Our revised manuscript:

Table 2. MAD of the optimized bond distances, angles, rotatable dihedrals in the gas phase using AIQM1, ANI-2x, and GFN2-xTB methods compared to the DFT (ω B97X-D/6-31G(d)) method for the QR50, PB20-QM-3k and PB20-QM-8k datasets.

	QR50 ^a			PB20-QM-3k ^a			PB20-QM-8k ^a	
	AIQM1 ^{b,c}	ANI-2x ^c	GFN2-xTB	AIQM1 ^{b,c}	ANI-2x ^c	GFN2-xTB	ANI-2x	GFN2-xTB
All systems		(50) ^d			(3156) ^d		(8776) ^d	
Bond (Å)	0.005	0.006	0.008	0.004	0.003	0.006	0.003	0.006
Angle (°)	0.6	0.9	0.8	0.4	0.5	0.6	0.6	0.7
Dihedral (°)	11.6	16.1	11.2	26.7	32.0	28.0	32.6	29.0
Neutral systems		(24) ^d			(3061) ^d		(7260) ^d	
Bond (Å)	0.004	0.003	0.007	0.004	0.003	0.006	0.003	0.006
Angle (°)	0.5	0.6	0.7	0.4	0.4	0.6	0.5	0.7
Dihedral (°)	10.6	12	10.6	27.2	32.5	28.6	35.2	31.6
Charged systems		(26) ^d			(95) ^d		(1516) ^d	
Bond (Å)	0.006	0.009	0.010	0.005	0.004	0.005	0.004	0.006
Angle (°)	0.7	1.3	0.9	0.6	0.8	0.8	0.8	0.8
Dihedral (°)	12.7	21	11.9	18.2	25.0	18.5	25.9	21.5

a. QR50: our selected 50 drugs/inhibitors (Fig. 1b and Supplementary Schemes. 1-3); PB20-QM-3k: a smallest subset dataset of PB20-QM containing 3156 drug/inhibitors (3125 molecules containing C, H, O and/or N elements, 15 molecules containing F, Cl and/or S elements, 16 molecules containing B, P, Se, Br and/or I elements); PB20-QM-8k: a smaller subset dataset of PB20-QM containing 8776 drug/inhibitors containing C, H, O, N, F, Cl and/or S elements. b. ONIOM(MLP:ANI-2x) method was used for the molecules containing F, Cl and/or S elements, when AIQM1 was used. c. ONIOM(MLP:SE) method was used for the molecules containing B, P, Se, Br and/or I elements, when AIQM1 or ANI-2x was used. d. Number of the drug/inhibitors

To further evaluate the accuracy of the MLPs methods for more different

functional groups, a new computational benchmark dataset taken from PDBbind v2020 (denoted as PB20-QM) containing 12963 drug/inhibitor molecules as well as its two smaller sub-datasets (PB20-QM-8k: 8k molecules containing, C, H, O, N, F, S, and/or Cl elements; and PB20-QM-3k: 3156 molecules mainly containing, C, H, O, and/or N elements) were also set up for geometry optimization by the different methods (Table 2). Compared to the reliable ω B97X-D method, our computational results show that MAD in the bond distances, angles and rotatable dihedrals of the drug/inhibitor molecules by the MLPs (AIQM1, ANI-2x) and SE (GFN2-xTB) methods vary by 0.003-0.006 Å, 0.4-0.6° and 26.7-32.0° respectively (Table 2).

In addition, the drug/inhibitor molecules containing charged group(s) were generally found to have larger structural deviations (e.g., MAD) when optimized by all MLPs and SE methods than the neutral drug/inhibitor molecules (Table 2), except the dihedrals in PB20-QM-3k possibly due to its scarce systems. For the neutral systems, these MLPs showed higher accuracy than GFN2-xTB with smaller structural deviation in bond (Δ MAD: 0.002-0.004 Å) and angle (Δ MAD: 0.1-0.2°) compared to the DFT method. The dihedral deviations for the AIQM1 and GFN2-xTB methods are generally comparable and smaller than those for MLP ANI-2x. Even the ANI-series MLPs were not primarily designed for charged systems, the ANI-2x method still showed a small structural deviation (MAD: <0.009 Å (bond), < 0.2° (angle)) for the charged systems. In comparison, the CC-level MLP AIQM1 method gives superior results to the ANI-series MLPs. Therefore, these MLPs, particularly the CC-quality AIQM1 method, can give reliable structures for drug/inhibitor systems at much lower computational costs. Moreover, apart from the charge effect, structural flexibility of the drug/inhibitor molecules is another key important factor in the larger RMSDs in gas phase (Supplementary Table 36).

Datasets setup. Our drug/inhibitor molecules downloaded from PDBbind v2020 dataset were used to set up a computational benchmark dataset (denoted as PB-

20-QM), which contains 12963 drug/inhibitor molecules after removing metal-containing systems. The protonation states of all these molecules remain preserved as those in the PDBbind v2020 dataset, except for minor corrections (to fix very poor or wrong positions of H atoms based on their corresponding PDB structures) to some molecules. Geometry optimization for all molecules in this PB-20-QM dataset in gas phase were conducted by the above-mentioned DFT and SE methods. In addition, two smaller sub-datasets (PB20-QM-8k: 8776 molecules containing, C, H, O, N, F, S, and/or Cl elements; and PB20-QM-3k: 3156 molecules mainly containing, C, H, O, and/or N elements) were also set up. Geometry optimization for all molecules in this PB20-QM-8k dataset were conducted by the above-mentioned DFT, SE and ANI-2x methods, whereas that in PB20-QM-3k dataset were performed by the above-mentioned DFT, SE, AIQM1, and ANI-2x methods.

Our revised supporting information 1:

Computations of the selected 50 systems (Supplementary Schemes 1-3) in gas phase

Supplementary Table 1 Correlation (R^2) and MAD of the refined bond distances, angles, rotatable dihedrals for the selected 50 drugs/inhibitors based on optimization in the gas phase using AIQM1, ANI-2x, and GFN2-xTB methods compared to the DFT (ω B97X-D/6-31G(d)) method. ONIOM(MLP:ANI-2x) method was used for the molecules containing F, Cl and/or S elements, when AIQM1 was used. ONIOM(MLP:SE) method was used for the molecules containing P and/or Br elements, when AIQM1 or ANI-2x was used.

		AIQM1	ANI-2x	GFN2-xTB
R^2	Bond	0.994	0.988	0.988
	Angle	0.977	0.916	0.968
	Rotatable Dihedral	0.953	0.932	0.955
MAD	Bond (Å)	0.005	0.006	0.008
	Angle (°)	0.6	0.9	0.8
	Rotatable Dihedral (°)	11.6	16.1	11.2

Supplementary Table 2 Correlation (R^2) and MAD of the refined bond distances, angles, rotatable dihedrals for neutral of the selected 50 drugs/inhibitors based on optimization in the gas phase using AIQM1, ANI-2x, and GFN2-xTB methods compared to the DFT (ω B97X-D/6-31G(d)) method. ONIOM(MLP:ANI-2x) method was used for the molecules containing F, Cl and/or S elements, when AIQM1 was used. ONIOM(MLP:SE) method was used for the molecules containing P and/or Br elements, when AIQM1 or ANI-2x was used.

		AIQM1	ANI-2x	GFN2-xTB
R²	Bond	0.995	0.996	0.990
	Angle	0.986	0.982	0.976
	Rotatable Dihedral	0.962	0.954	0.961
MAD	Bond (Å)	0.004	0.003	0.007
	Angle (°)	0.5	0.6	0.7
	Rotatable Dihedral (°)	10.6	12.0	10.6

Supplementary Table 3 Correlation (R^2) and MAD of the refined bond distances, angles, rotatable dihedrals for charged of the selected 50 drugs/inhibitors based on optimization in the gas phase using AIQM1, ANI-2x, and GFN2-xTB methods compared to the DFT (ω B97X-D/6-31G(d)) method. ONIOM(MLP:ANI-2x) method was used for the molecules containing F, Cl and/or S elements, when AIQM1 was used. ONIOM(MLP:SE) method was used for the molecules containing P and/or Br elements, when AIQM1 or ANI-2x was used.

		AIQM1	ANI-2x	GFN2-xTB
R²	Bond	0.992	0.979	0.985
	Angle	0.966	0.826	0.957
	Rotatable Dihedral	0.943	0.907	0.949
MAD	Bond (Å)	0.006	0.009	0.010
	Angle (°)	0.7	1.3	0.9
	Rotatable Dihedral (°)	12.7	21.0	11.9

Supplementary Figure 1. Violin plots of deviation in (a) bond distances, (b) angles and (c) rotatable dihedrals for the 3156 drug/inhibitor structures (PB20-QM-3k: taken from our PB20-QM dataset) optimized in the gas phase, using MLPs (MLP-CC: AIQM1, MLP-DFT: ANI-2x) and the (SE) GFN2-xTB method compared to the (QM) ω B97X-D/6-31G(d) method. ONIOM(MLP:ANI-2x) method was used for the 15 molecules containing F, Cl and/or S elements, when AIQM1 was used. ONIOM(MLP:SE) method was used for the 16 molecules containing B, P, Se, Br and/or I elements, when AIQM1 or ANI-2x was used.

Supplementary Table 4 Correlation (R^2) and MAD of the refined bond distances, angles, rotatable dihedrals for the 3156 drug/inhibitor structures (PB20-QM-3k: taken from our PB20-QM dataset) optimized in the gas phase, using MLPs (MLP-CC: AIQM1, MLP-DFT: ANI-2x) and the (SE) GFN2-xTB method compared to the (QM) ω B97X-D/6-31G(d) method. ONIOM(MLP:ANI-2x) method was used for the molecules containing F, Cl and/or S elements, when AIQM1 was used. ONIOM(MLP:SE) method was used for the molecules containing B, P, Se, Br and/or I elements, when AIQM1 or ANI-2x was used.

		AIQM1	ANI-2x	GFN2-xTB
R²	Bond	0.999	0.999	0.998
	Angle	0.991	0.989	0.982
	Rotatable Dihedral	0.421	0.304	0.405
MAD	Bond (Å)	0.004	0.003	0.006
	Angle (°)	0.4	0.5	0.6
	Rotatable Dihedral (°)	26.7	32.0	28.0

Supplementary Table 5 Correlation (R^2) and MAD of the refined bond distances, angles, rotatable dihedrals for neutral of the 3156 drug/inhibitor structures (PB20-QM-3k: taken from our PB20-QM dataset) optimized in the gas phase, using MLPs (MLP-CC: AIQM1, MLP-DFT: ANI-2x) and the (SE) GFN2-xTB method compared to the (QM) ω B97X-D/6-31G(d) method. ONIOM(MLP:ANI-2x) method was used for the molecules containing F, Cl and/or S elements, when AIQM1 was used. ONIOM(MLP:SE) method was used for the molecules containing B, P, Se, Br and/or I elements, when AIQM1 or ANI-2x was used.

		AIQM1	ANI-2x	GFN2-xTB
R²	Bond	0.999	0.999	0.998
	Angle	0.992	0.989	0.983
	Rotatable Dihedral	0.409	0.293	0.391
MAD	Bond (Å)	0.004	0.003	0.006
	Angle (°)	0.4	0.4	0.6
	Rotatable Dihedral (°)	27.2	32.5	28.6

Supplementary Table 6 Correlation (R^2) and MAD of the refined bond distances, angles, rotatable dihedrals for charged of the 3156 drug/inhibitor structures (PB20-QM-3k: taken from our PB20-QM dataset) optimized in the gas phase, using MLPs (MLP-CC: AIQM1, MLP-DFT: ANI-2x) and the (SE) GFN2-xTB method compared to the (QM) ω B97X-D/6-31G(d) method. ONIOM(MLP:ANI-2x) method was used for the molecules containing F, Cl and/or S elements, when AIQM1 was used. ONIOM(MLP:SE) method was used for the molecules containing B, P, Se, Br and/or I elements, when AIQM1 or ANI-2x was used.

		AIQM1	ANI-2x	GFN2-xTB
R²	Bond	0.999	0.999	0.998
	Angle	0.980	0.964	0.966
	Rotatable Dihedral	0.629	0.487	0.634
MAD	Bond (Å)	0.005	0.004	0.005
	Angle (°)	0.6	0.8	0.8
	Rotatable Dihedral (°)	18.2	25.0	18.5

Supplementary Figure 2. Violin plots of deviation in (a) bond distances, (b) angles and (c) rotatable dihedrals for the 8776 drug/inhibitor structures (PB20-QM-8k: taken from our PB20-QM dataset) optimized in the gas phase, using ANI-2x and the (SE) GFN2-xTB method compared to the (QM) ω B97X-D/6-31G(d) method.

Supplementary Table 7 Correlation (R^2) and MAD of the refined bond distances, angles, rotatable dihedrals for the 8776 drug/inhibitor structures (PB20-QM-8k: taken from our PB20-QM dataset) optimized in the gas phase, using ANI-2x and the (SE) GFN2-xTB method compared to the (QM) ω B97X-D/6-31G(d) method.

		ANI-2x	GFN2-xTB
R^2	Bond	0.999	0.998
	Angle	0.981	0.977
	Rotatable Dihedral	0.316	0.401
MAD	Bond (Å)	0.003	0.006
	Angle (°)	0.6	0.7
	Rotatable Dihedral (°)	32.6	29.0

Supplementary Table 8 Correlation (R^2) and MAD of the refined bond distances, angles, rotatable dihedrals for neutral of the 8776 drug/inhibitor structures (PB20-QM-8k: taken from our PB20-QM dataset) optimized in the gas phase, using ANI-2x and the (SE) GFN2-xTB method compared to the (QM) ω B97X-D/6-31G(d) method.

		ANI-2x	GFN2-xTB
R^2	Bond	0.999	0.998
	Angle	0.985	0.978
	Rotatable Dihedral	0.261	0.347
MAD	Bond (Å)	0.003	0.006
	Angle (°)	0.5	0.7
	Rotatable Dihedral (°)	35.2	31.6

Supplementary Table 9 Correlation (R^2) and MAD of the refined bond distances, angles, rotatable dihedrals for charged of the 8776 drug/inhibitor structures (PB20-QM-8k: taken from our PB20-QM dataset) optimized in the gas phase, using ANI-2x and the (SE) GFN2-xTB method compared to the (QM) ω B97X-D/6-31G(d) method.

		ANI-2x	GFN2-xTB
R^2	Bond	0.998	0.998
	Angle	0.963	0.968
	Rotatable Dihedral	0.480	0.565
MAD	Bond (Å)	0.004	0.006
	Angle (°)	0.8	0.8
	Rotatable Dihedral (°)	25.9	21.5

Comment 2: The chemical space covered by the test cases is an issue. Taking into account the vast chemical space covered by drug-protein complexes, the authors should explain how the current set of complexes (or in light of the previous point, the new set of benchmark complexes) covers at least the most frequent functional groups in drug discovery. In addition, the authors should also explain the reason for not including any phosphorus-containing drug in their study.

Our Response: Thanks again for your insightful comment. We also agree with your point. First, our new computational benchmark dataset (PB20-QM), which has been mentioned in our above response 1, covers much more different functional groups. In addition, based on one recent review about the 40 most frequent functional groups (“The Most Common Functional Groups in Bioactive Molecules and How Their Popularity Has Evolved over Time”, *J. Med. Chem.* **2020**, 8408), we have further used these functional groups to search 40 more protein-drug/inhibitor systems from the protein databank and to perform new multiscale quantum refinements. As shown in Supplementary Figure 1 and Schemes 1-3 (please see below), all drugs from our previous and latest systems have almost met all most frequent functional groups in this review papers except for very limited and not-good-quality crystal structures for 3 very less common functional groups (Supplementary Table 1). We have tried very hard to search and expand more systems, but it should be noted that it is quite time-consuming to select and check appropriate systems as well as set up/prepare these protein systems before our multiscale quantum refinement simulations.

Moreover, the reason not including any phosphorus-containing drug in our previous study attributes to the fact that current ANI-2x or AIQM1 MLPs cannot apply for the P element. To address your comment, we have added 2 new protein systems containing phosphorus-containing drug/inhibitor molecules (Supplementary Scheme 3), in which the combination of MLPs and xTB have been further applied to describe these drug molecules. The major and core molecules are treated by MLPs, whereas the remaining drug part containing P element are treated by xTB method. We have added these new results, refs. S1 & S1, and discussion in our revised manuscript and key ones in our revised SI (extensively detailed raw data can be found in pages S88-S304) as

follow:

Our Action:

Our revised manuscript:

(Updated) Fig. 3: Violin plots of deviation in (a) bond distances, (b) angles and (c) rotatable dihedrals for the selected 50 drug/inhibitor structures (QR50 dataset) optimized in the gas phase, using MLPs (MLP-CC: AIQM1, MLP-DFT: ANI-2x) and the (SE) GFN2-xTB method compared to the (QM) ω B97X-D/6-31G(d) method. ONIOM(MLP:ANI-2x) method was used for the molecules containing F, Cl and/or S elements, when AIQM1 was used. ONIOM(MLP:SE) method was used for the molecules containing P and Br elements, when AIQM1 or ANI-2x was used.

To further evaluate the accuracy of the MLPs methods for more different functional groups,⁵¹

51. Ertl P., Altmann E., McKenna J. M. The Most Common Functional Groups in Bioactive Molecules and How Their Popularity Has Evolved over Time. *J. Med. Chem.* **63**, 8408-8418 (2020).

Table 2. MAD of the optimized bond distances, angles, rotatable dihedrals in the gas phase using AIQM1, ANI-2x, and GFN2-xTB methods compared to the DFT (ω B97X-D/6-31G(d)) method for the QR50, PB20-QM-3k and PB20-QM-8k datasets.

	QR50 ^a			PB20-QM-3k ^a			PB20-QM-8k	
	AIQM1 ^{b,c}	ANI-2x ^c	GFN2-xTB	AIQM1 ^{b,c}	ANI-2x ^c	GFN2-xTB	ANI-2x	GFN2-xTB
All systems		(50) ^d			(3156) ^d			(8776) ^d
Bond (Å)	0.005	0.006	0.008	0.004	0.003	0.006	0.003	0.006
Angle (°)	0.6	0.9	0.8	0.4	0.5	0.6	0.6	0.7
Dihedral (°)	11.6	16.1	11.2	26.7	32.0	28.0	32.6	29.0
Neutral systems		(24) ^d			(3061) ^d			(7260) ^d
Bond (Å)	0.004	0.003	0.007	0.004	0.003	0.006	0.003	0.006
Angle (°)	0.5	0.6	0.7	0.4	0.4	0.6	0.5	0.7
Dihedral (°)	10.6	12	10.6	27.2	32.5	28.6	35.2	31.6
Charged systems		(26) ^d			(95) ^d			(1516) ^d
Bond (Å)	0.006	0.009	0.010	0.005	0.004	0.005	0.004	0.006
Angle (°)	0.7	1.3	0.9	0.6	0.8	0.8	0.8	0.8
Dihedral (°)	12.7	21	11.9	18.2	25.0	18.5	25.9	21.5

a. QR50: our selected 50 drugs/inhibitors (Fig. 1b and Supplementary Schemes. 1-3); PB20-QM-3k: a smallest subset dataset of PB20-QM containing 3156 drug/inhibitors (3125 molecules containing C, H, O and/or N elements, 15 molecules containing F, Cl and/or S elements, 16 molecules containing B, P, Se, Br and/or I elements); PB20-QM-8k: a smaller subset dataset of PB20-QM containing 8776 drug/inhibitors containing C, H, O, N, F, Cl and/or S elements. b. ONIOM(MLP:ANI-2x) method was used for the molecules containing F, Cl and/or S elements, when AIQM1 was used. c. ONIOM(MLP:SE) method was used for the molecules containing B, P, Se, Br and/or I elements, when AIQM1 or ANI-2x was used. d. Number of the drug/inhibitors

(Updated) Fig. 3: Box plots of deviation of (a) RSZD scores as well as (b) strain energy ($\Delta\Delta E$, kcal/mol) and violin plots of the deviations of (c) bond distances, (d) angles, and (e) rotatable dihedrals of drugs/inhibitors in the selected 50 protein–drug/inhibitor

systems after **M1-M10** quantum refinement approaches compared to **M7** (ONIOM3(DFT:SE:MM)). The X-ray results were taken from the experimental structures without further refinement.

Quantum refinement of protein–drug/inhibitor systems. These MLPs 50 protein–drug/inhibitor systems (Fig. 1 and Supplementary Schemes 1-3). acylated ceftazidime and isatin, including its covalent linkages) after various QRs were reduced by 1.0-1.1 on average (Fig. 3a and Supplementary Table 22), isatin for DJ-1 system, whose RSZD scores can be significantly decreased from 7.6/6.0 (X-ray crystal structure) to 0.5-1.0/0.9-2.1

In addition, the computed strain energies (Fig. 3b) of these 50 drugs/inhibitors were considerably reduced by 27.1~31.4 kcal/mol in the taurocholic acid-CmeR (from 239.8 to approximately 78.7-85.6 kcal/mol) and attributed to the underestimated C-S bond (b5: 0.18-0.22 Å shorter) of taurocholic acid and two C-C bonds (b₃ and b₄: 0.25~0.28 Å shorter) of darunavir in the X-ray structures (Fig. 1b, Supplementary Fig. 68, and Supplementary Table 15)

Combined MLP-CC/SE methods for broader drug molecules. Moreover, the higher-level AIQM1 and GFN2-xTB methods were further combined through the ONIOM scheme for our QRs on the three selected protein–drug/inhibitor systems (Supplementary Scheme 3), in which the major core of the drug/inhibitor structures and their remaining parts containing other elements (P or Br) are described by the AIQM1 and SE methods, respectively. The RMSD and MAD for these refined drug/inhibitor structures with reference to ONIOM3(DFT:SE:MM) (**M7**) were very small (RMSD: bonds < 0.011 Å, angles < 0.7°, dihedrals < 2.6°; MAD: bonds < 0.011 Å, angles < 0.5°, dihedrals < 2.3°). Moreover, our refined drug/inhibitors structures can give similar electron density around the binding site to the reliable **M7** scheme (Supplementary Figs. 176, 181 and 219). What's more, their RSZD score were reduced from 1.4-3.9 (X-ray crystal structure) to 1.3-2.5 after these QRs. Therefore, these results show that the

combined MLP-CC with SE method can further resolve limitations of MLPs and be applied to much more elements and molecules.

Fig. 5: Correlation (R^2) of the refined bond distances, angles, rotatable dihedrals, RSZD scores and strain energy for the (a) neutral and (b) charged drugs/inhibitors obtained by the **M8**, **M10** and **M6R** schemes (ONIOM3(ANI-2x:SE:MM), ONIOM3(AIQM1:SE:MM) and ONIOM2(SE:MM), respectively) compared to those obtained by the **M7** scheme (ONIOM3(DFT:SE:MM)).

Correlation. Pleasingly, for 39 of the 50 selected systems (Supplementary Tables 4 and 22), charged and neutral drug/inhibitor systems (bond distances: $R^2 > 0.988$; angles: $R^2 > 0.962$)

For neutral systems, SE-based (ONIOM2(SE:MM), **M6R**) schemes gave RSZD scores ($R^2 > 0.971$) **M8**, **M10** and **M6R** schemes also showed good agreement for the dihedrals ($R^2 > 0.935$) those of the DFT-based scheme ($R^2 \sim 0.996$ and ~ 0.992 , respectively), significantly better than the those of SE-based scheme (**M6R**,

$R^2 \sim 0.910$) gave the lowest R^2 values for the angles (0.962), dihedrals (0.924) and strain energy (0.869) DFT-based scheme (M7; $R^2 \sim 0.995-0.996$), as well as the SE-based scheme (M6R, $R^2 \sim 0.996$). Promisingly, the AIQM1-based scheme (M10) showed outstanding R^2 on the bonds (~ 0.995) and dihedrals (~ 0.974)

To further apply MLPs to even boarder drug/inhibitor molecules containing P or Br element, AIQM1 (or ANI-2x) and SE methods were also combined to describe the drug/inhibitor molecules by the ONIOM approach (eq 5, which replaces the $E_{high,model}$ part in ONIOM2(QM:MM) in eq 2 or that in ONIOM3(QM:SE:MM) in eq 3).

$$E_{ONIOM2(MLP:xTB)} = E_{MLP,model} + E_{xTB,real} - E_{xTB,model} \quad (5)$$

Our revised supporting information 1:

1. General computational details

In order to evaluate the performance and reliability of a few MLPs in our quantum refinements, 50 drug/inhibitor molecules containing only H, C, N, O, F, S, Cl, P, Br elements were selected (Supplementary Scheme 1). 27 of these molecules comprise only H, C, N, O elements, 20 molecules contain H, C, N, O, F, S, and/or Cl elements, and 3 molecules contain H, C, N, O, S, Br, P elements.

{H, C, N, O}

Supplementary Scheme 1. Chemical structures of our selected 27 drug/inhibitor molecules containing H, C, N, O elements in this study. Their drug name, related target protein name and PDB ID are included beneath these structures.

Supplementary Scheme 1 (Continued) Chemical structures of our selected 27 drug/inhibitor molecules containing H, C, N, O elements in this study. Their drug name, related target protein name and PDB ID are included beneath these structures.

{H, C, N, O, S, F, Cl}

Supplementary Scheme 2. Chemical structures of our selected 20 drug/inhibitor molecules containing H, C, N, O, F, S, and/or Cl elements in this study. Their drug name, related target protein name and PDB ID are included beneath these structures. Dashed lines represent the ONIOM(MLP-CC:MLP-DFT) boundary within the drug/inhibitor molecules containing F, Cl and/or S elements, when AIQM1 or ANI-1ccx was used. The region containing only C, H, O, N described by AIQM1 or ANI-1ccx, where the rest by ANI-2x.

Supplementary Scheme 3. Chemical structures of our 3 selected drug/inhibitor molecules containing H, C, N, O, S, Br, and/or P elements in this study. Their drug name, related target protein name and PDB ID are included beneath these structures. Dashed lines represent the ONIOM(MLP:SE) boundary within the drug/inhibitor molecules containing P and/or Br elements, when AIQM1 or ANI-2x was used. The region containing only C, H, O, N described by AIQM1 or ANI-1ccx, where the rest containing P or Br by GFN2-xTB.

FG1 40.2	FG2 38.8	FG3 29.5	FG4 23.1	FG5 19.6	FG6 19.6	FG7 13.6	FG8 13.1
FG9 12.2	FG10 10.3	FG11 9.4	FG12 6.5	FG13 6.2	FG14 5.4	FG15 5.2	FG16 5.2
FG17 5.0	FG18 4.9	FG19 4.9	FG20 3.0	FG21 3.7	FG22 3.1	FG23 2.6	FG24 2.1
FG25 3.0	FG26 1.9	FG27 1.8	FG28 1.5	FG29 1.3	FG30 1.2	FG31 0.8	FG32 0.6
FG33 0.6	FG34 0.6	FG35 0.6	FG36 0.5	FG37 0.5	FG38 0.5	FG39 0.4	FG40 0.4

Supplementary Figure 4 The 40 most frequent functional groups occurring in bioactive molecules described reported and adapted from a recent paper.^{S1} The number indicates the percentage of molecules containing this functional group. R indicates aliphatic or aromatic carbon (Adapted with permission from ref S1. Copyright 2020 American Chemical Society.).

S1. Ertl P., Altmann E., McKenna J. M. The Most Common Functional Groups in Bioactive Molecules and How Their Popularity Has Evolved over Time. *J. Med. Chem.* **63**, 8408-8418 (2020).

Supplementary Table 10 The 40 most frequent functional groups (FGs, Supplementary Figure 1) occurring in our 50 selected drug/inhibitor molecules in this study.

FGs	PDB ID	FGs	PDB ID	FGs	PDB ID
FG1	1XBB, 5P9I, 4HZZ, 4YHM, 6JX4, 7RF3, 1BR5, 1D4I, 1LF3, 1OWK, 1PF7, 1XOZ, 2WCA, 7RFW, 2W26, 5HLS, 3QQA, 6V83, 2AX6, 3EQ7, 4ZB8, 5L7I	FG2	5P9I, 4HZZ, 4XH6, 6JX4, 1AJX, 1D4I, 1LF3, 1YAT, 2P7Z, 2WCA, 2WI1, 1NJ1, 1PA9, 2QBR, 5Y62, 1UYG	FG3	1XBB, 4YHM, 6JX4, 2P7Z, 3FR2, 6PUB, 4KRA, 1NOX, 1PA9, 1M7Q
FG4	7RFW, 4KRA, 2AX6, 3EQ7, 1M7Q, 1UYG	FG5	1XBB, 4YHM, 6JX4, 7RF3, 1OWK, 1PF7, 1XOZ, 3ATM, 4KRA, 2QBR, 5CT1, 1M7Q	FG6	2W26, 5HLS, 4P6W, 2QBR, 5L7I, 1M7Q, 2IWU
FG7	1AJX, 1BR5, 1D4I, 1FG9, 1LF3, 1PF7, 1YAT, 2A3I, 2WCA, 5KR1, 3QQA, 1NJ1, 1NOX, 2AX6, 5Y62	FG8	4XH6, 1BR5, 1PF7, 4KRA	FG9	4HZZ, 4YHM, 1P1O, 2HXM, 3FR2, 4KRA, 6V83, 2QBR, 4ZB8
FG10	2P7Z, 5Y1Y, 1PA9, 2IWU	FG11	5KR1	FG12	5P9I, 1UY9, 2WI1, 5KR1, 6V83, 1NJ1, 2QBR, 2UYG
FG13	4HZZ, 1P1O, 3ATM, 1NJ1, 4ZB8	FG14	1AJX	FG15	1FG9, 1YAT, 4P6W, 2IWU
FG16	7RFW	FG17	2A3I, 4P6W, 2IWU	FG18	1YAT, 2P7Z, 6V83, 2IWU
FG19	2W26, 6V83, 1PA9, 2QBR	FG20	2WCA, 2W26, 5KR1	FG21	2QBR
FG22	5L7I, 3QQA, 1PA9, 5CT1	FG23	1BR5, 5Y62	FG24	5Y1Y, 1PA9, 2AX6
FG25	None	FG26	None	FG27	1LF3, 1UY9, 1XOZ, 5KR1
FG28	4YHM, 1OWK, 1UY9, 1NJ1, 5Y62, 1UYG	FG29	6JX4, 5P9I	FG30	6PUB
FG31	None	FG32	4ZB8	FG33	1NJ1
FG34	4XH6, 2A3I, 4P6W, 4KRA	FG35	1NJ1	FG36	1NOX, 5Y62
FG37	5HLS, 6V83	FG38	1YAT, 6AFA	FG39	1F9G
FG40	1LF3, 3EQ7, 1M7Q				

Computations of the selected 50 protein-drug/inhibitor systems (Supplementary Schemes 1-3)

After tests on selected 10 protein-drug/inhibitor systems (shown in Fig. 1), the CC-level AIQM1 based methods (**M5**, **M5a**, **M10** and **M10a**) yields better results than

the other CC-level ANI-1ccx based methods (**M4**, **M4a**, **M9** and **M9a**). Therefore, we choose AIQM1 as the only one CC-level method for the following quantum refinements. ONIOM(AIQM1:ANI-2x) based method (**M5a** and **M10a**) were used to describe the drug/inhibitor molecules (C, H, O, N, F, Cl, S) due to the element limitation (C, H, O, N) of AIQM1. For the three drug/inhibitor molecules (C, H, O, N, F, Cl, S, Br, P) in 2QBR, 1NOX and 5Y62, ONIOM(AIQM1: GFN2-xTB) were used due to the element limitation (C, H, O, N) of AIQM1.

Supplementary Table 11 Correlation (R^2) and MAD of the refined bond distances, angles, rotatable dihedrals for all drugs/inhibitors in the selected 50 protein-drug/inhibitor systems from **M1-M10** QRs with respect to **M7** as the reference. Those results for X-ray were taken from the experimental structure without our further refinement.

		M1	M3	M5	M6	M8	M10	M6R	X-ray
R²	Bond	0.995	0.990	0.991	0.986	0.993	0.996	0.991	0.843
	Angle	0.990	0.978	0.985	0.980	0.981	0.992	0.985	0.717
	Rotatable Dihedral	0.999	0.999	0.999	0.999	1.000	1.000	1.000	0.992
MAD	Bond (Å)	0.005	0.007	0.007	0.009	0.005	0.004	0.007	0.027
	Angle (°)	0.4	0.7	0.6	0.7	0.6	0.4	0.6	2.5
	Rotatable Dihedral (°)	1.2	1.7	1.6	1.5	1.2	1.0	1.1	6.6

Supplementary Table 12 Correlation (R^2) and MAD of the refined bond distances, angles, rotatable dihedrals for the neutral drugs/inhibitors in the selected 50 protein-drug/inhibitor systems from **M1-M10** QRs with respect to **M7** as the reference. Those results for X-ray were taken from the experimental structure without our further refinement.

		M1	M3	M5	M6	M8	M10	M6R	X-ray
R²	Bond	0.997	0.995	0.994	0.992	0.998	0.997	0.991	0.841
	Angle	0.992	0.989	0.989	0.985	0.995	0.995	0.982	0.755
	Rotatable Dihedral	0.999	0.999	0.999	1.000	1.000	1.000	1.000	0.990
MAD	Bond (Å)	0.004	0.005	0.005	0.007	0.003	0.004	0.007	0.024
	Angle (°)	0.4	0.5	0.5	0.6	0.4	0.3	0.6	2.3
	Rotatable Dihedral (°)	1.1	1.3	1.3	1.3	0.8	0.9	1.2	7.0

Supplementary Table 13 Correlation (R^2) and MAD of the refined bond distances, angles, rotatable dihedrals for charged drugs/inhibitors in selected 50 protein-drug/inhibitor systems from **M1-M10** QRs with respect to **M7** as the reference. Those results for X-ray were taken from the experimental structure without our further refinement.

		M1	M3	M5	M6	M8	M10	M6R	X-ray
R²	Bond	0.993	0.983	0.986	0.978	0.988	0.995	0.991	0.846
	Angle	0.987	0.964	0.979	0.973	0.962	0.987	0.988	0.661
	Rotatable Dihedral	1.000	0.999	0.999	0.999	0.999	1.000	1.000	0.994
MAD	Bond (Å)	0.006	0.009	0.008	0.011	0.008	0.005	0.008	0.030
	Angle (°)	0.5	0.9	0.7	0.8	0.8	0.5	0.6	2.7
	Rotatable Dihedral (°)	1.3	2.1	2.0	1.8	1.6	1.2	0.9	6.2

Supplementary Figure 5 Correlation (R^2) of the refined bond distances, angles, rotatable dihedrals, RSZD score and strain energy for our (a) neutral and (b) charged

drugs/inhibitors in the all 50 protein-drug/inhibitor systems from **M1-M10** QRs with respect to **M7** as the reference.

Supplementary Table 14 RSZD scores of the all 50 drugs/inhibitors in the proteins from our quantum refinement calculations (**M1-M10**). Those results for X-ray were taken from the experimental structure without our further refinement.

PDB ID	M1	M3	M5	M6	M7	M8	M10	M6R	X-ray
1xbb	3.9	3.7	3.5	3.7	3.9	3.8	3.5	3.8	4.1
2w26	0.3	0.3	0.4	0.3	0.5	0.4	0.3	0.3	1.1
4hzz	1.3	1.3	1.3	1.5	1.4	1.4	1.5	1.6	2.8
4kra	1.9	1.2	1.7	1.2	1.4	1.5	1.2	1.1	2.7
4p6w	5.6	5.6	5.5	5.4	5.6	5.5	5.5	5.2	7.7
5hls	0.8	0.9	0.8	0.7	0.7	0.6	0.6	0.8	1.3
5kr1	4.4	4.3	4.2	4.5	4.3	4.3	4.2	4.6	5.0
5p9i	1.8	1.8	1.7	1.8	1.7	1.8	1.8	1.8	1.9
6jx4	0.4	0.4	0.4	0.5	0.4	0.5	0.4	0.5	1.6
7rfw^a	0.7	0.9	1.0	0.8	0.5	0.6	0.7	0.5	7.6
3qqa	2.8	2.8	2.8	2.8	2.8	2.8	2.8	2.8	2.7
4xh6	4.3	4.3	4.3	4.2	4.3	4.2	4.2	4.1	4.3
4yhm	3.1	3.1	3.0	3.1	3.1	3.1	3.1	3.1	3.8
6v83^a	2.0	1.8	1.7	1.9	2.1	2.3	2.3	1.9	2.0
1ajx	1.1	1.1	1.1	1.0	1.0	1.1	1.0	0.9	1.9
1br5	0.7	0.7	0.7	0.9	0.9	0.8	0.7	0.7	0.6
1d4i	0.2	0.1	0.2	0.1	0.2	0.1	0.1	0.3	2.2
1f9g	0.3	0.4	0.3	0.3	0.3	0.4	0.3	0.3	0.4
1lf3	1.9	1.9	1.9	1.9	1.9	1.9	1.9	2.0	2.3
4zb8	1.9	1.9	1.9	1.9	1.6	1.7	1.6	1.7	3.7
1p1o	2.3	2.3	2.3	2.5	2.4	2.5	2.5	2.6	2.4
1pf7	0.4	0.4	0.8	0.4	0.3	0.3	0.3	0.4	1.3
1m7q	8.4	8.5	8.5	8.5	8.4	8.5	8.5	8.5	9.6
1pa9	0.3	0.2	0.3	0.6	0.4	0.3	0.2	0.3	2.3
1xoz	0.0	0.0	0.0	0.0	0.0	0.0	0.0	0.0	3.6
2ax6	1.6	1.6	1.6	1.6	1.6	1.6	1.6	1.6	1.8
2iwu	2.5	2.5	2.5	2.6	2.5	2.5	2.5	2.6	2.5
2p7z	0.9	0.9	0.8	0.9	0.5	0.8	0.5	0.5	1.8
1uy9	0.1	0.1	0.1	0.1	0.1	0.1	0.1	0.1	0.3
3eq7	2.1	2.2	2.1	2.0	2.1	2.1	2.1	2.0	2.4
5y1y	1.1	1	1.1	1.3	1.3	1.2	1.1	1.2	1.2
5y62	1.2	1.2	1.3	1.2	1.3	1.3	1.3	1.3	1.4
1nox	1.7	1.5	1.5	1.6	1.6	1.3	1.5	1.6	2.5
1nj1	2.5	2.5	2.3	2.1	2.0	2.2	2.3	1.7	3.9
2a3i	1.1	1.1	1.1	1.1	1.2	1.1	1.1	1.1	1.6
5ct1	2.1	2.1	2.2	2.1	2.1	2.1	2.1	2.1	2.1

6afa^a	2.2	2.1	0.9	2.0	0.8	2.1	1.0	1.1	6.0
6pub	0.9	1.3	1.2	1.1	0.9	0.9	0.8	0.9	2.2
1owk	1.4	1.4	1.4	1.4	1.5	1.4	1.4	1.4	1.7
2qbr	2.4	2.4	2.4	2.4	2.6	2.5	2.6	2.7	3.9
1yat	2.6	2.6	2.5	2.6	2.6	2.5	2.5	2.6	3.1
1xp9	0.4	0.4	0.4	0.4	0.4	0.4	0.4	0.4	1.2
2hxm	9.4	9.5	9.2	9.4	9.3	9.4	9.1	9.2	10.4
2wca	1.1	1.1	1.1	1.2	0.6	0.6	0.6	0.6	2.5
2wi4	1.5	1.6	1.7	1.8	1.5	1.5	1.6	1.7	3.5
1uyg	0.0	0.0	0.0	0.0	0.0	0.0	0.0	0.0	0.6
2wi1	2.1	2.1	2.1	2.1	2.1	2.0	2.0	2.1	4.4
3fr2	0.7	0.5	0.7	0.7	0.8	0.5	0.4	0.5	0.8
3atm	1.8	1.8	1.8	1.9	1.8	1.8	1.8	1.8	2.1
5l7i	1.2	1.2	1.3	1.5	1.2	1.2	1.2	1.5	2.8
Average	1.9	1.9	1.9	1.9	1.9	1.9	1.8	1.8	2.9

a. The RSZD scores contributed from the drug/inhibitor and its covalent linkage residue.

Supplementary Table 15 Strain energy (ΔE , kcal/mol) at ω B97X-D/6-31G(d) level of the all 50 drugs/inhibitors in the proteins from our quantum refinement calculations (M1-M10). Those results for X-ray were taken from the experimental structure without our further refinement.

PDB ID	M1	M3	M5	M6	M7	M8	M10	M6R	X-ray
1xbb	18.29	17.28	15.22	18.79	17.88	17.66	15.83	20.87	40.40
4hzz	19.75	34.66	21.72	22.69	22.61	42.44	24.20	26.94	96.31
5p9i	5.32	5.29	4.03	7.60	4.99	5.42	4.01	12.92	12.90
6jx4	60.71	63.38	62.51	62.56	60.99	64.24	63.54	61.48	108.58
2w26	13.42	14.93	15.33	17.22	15.05	15.07	15.02	17.70	37.35
4kra	46.22	52.29	49.36	49.66	55.31	58.46	58.51	59.55	58.53
4p6w	7.07	8.83	8.88	12.60	6.95	8.38	8.41	12.29	57.51
5hls	8.46	9.76	9.32	10.79	8.41	9.13	10.47	10.77	12.83
5kr1	15.67	16.11	16.07	20.41	14.08	15.92	16.31	19.97	117.26
7rfw^a	29.24	32.40	32.57	34.90	29.49	32.05	33.36	34.61	60.94
1ajx	17.94	18.76	18.55	19.98	17.97	20.42	19.08	21.10	46.10
1br5	19.00	19.72	19.51	19.98	18.81	19.61	19.86	22.60	45.38
1d4i	29.19	30.42	30.96	33.41	31.45	32.00	32.67	34.52	41.89
1f9g	10.94	15.68	12.40	12.34	9.53	15.26	11.56	11.40	37.76
1lf3	57.16	57.63	59.73	61.74	57.27	57.75	60.36	62.08	121.07
1m7q	15.41	18.17	16.69	18.97	16.05	19.36	17.81	19.60	99.64
1nj1	29.19	34.15	37.02	38.79	42.22	35.16	35.26	47.50	40.91
1nox	14.17	16.20	16.00	16.18	14.97	15.37	15.41	16.19	79.05
1owk	22.12	32.99	23.27	27.27	17.42	25.53	18.93	21.54	22.11
1p1o	26.27	28.95	26.95	28.07	35.75	47.42	37.09	37.84	34.55

1pa9	5.53	9.39	6.51	7.71	7.11	12.00	9.74	8.60	44.48
1pf7	34.85	37.80	35.95	36.43	34.50	37.56	35.45	36.26	55.30
1uy9	2.38	2.87	3.10	4.55	2.43	2.59	2.53	4.87	18.96
1uyg	8.27	8.59	8.48	9.26	8.99	9.79	9.37	10.55	46.12
1xoz	3.28	3.93	3.87	4.10	3.40	3.93	4.02	4.46	21.23
1xp9	44.01	48.54	48.26	48.54	45.52	52.61	49.88	50.41	88.32
1yat	31.33	32.63	32.32	34.08	31.44	32.80	32.72	34.94	65.54
2a3i	12.52	12.45	12.81	14.31	11.06	11.68	12.25	15.85	42.33
2ax6	3.03	3.38	3.14	4.85	3.93	3.62	3.88	6.10	56.67
2hxm	5.45	12.82	7.08	9.76	8.02	13.29	9.51	13.14	99.07
2iwu	21.42	23.97	22.38	22.16	21.21	23.79	22.48	22.36	54.00
2p7z	15.48	18.53	17.98	17.97	12.29	18.01	14.92	15.82	33.00
2qbr	15.37	18.91	18.73	19.16	16.26	28.81	20.27	18.23	40.73
2wca	16.08	16.99	16.55	18.02	16.25	16.85	16.56	18.08	17.34
2wi1	13.01	14.06	13.75	14.63	13.30	13.25	13.45	14.56	34.29
2wi4	8.22	8.60	9.43	11.85	7.52	7.07	8.91	11.41	18.49
3atm	1.69	3.48	2.28	2.83	2.49	5.04	2.78	4.29	5.71
3eq7	16.63	16.81	16.77	18.74	16.77	16.79	16.65	18.07	37.57
3fr2	10.86	13.46	11.35	12.70	10.85	14.29	11.68	13.01	38.13
3qqa	79.14	85.57	78.66	84.73	80.66	83.98	83.48	84.00	239.82
4xh6	15.51	18.15	18.09	17.84	16.20	17.36	17.82	17.97	25.63
4yhm	41.56	48.48	43.31	48.80	39.96	48.41	41.71	43.10	61.92
4zb8	66.89	75.81	71.30	73.65	73.62	82.89	77.53	77.67	89.45
5ct1	37.27	41.56	42.13	38.84	38.06	43.72	43.00	39.56	59.43
5y1y	5.50	7.40	5.92	6.42	6.61	8.05	7.58	7.59	31.86
5y62	37.06	38.47	38.47	42.24	34.54	36.38	36.78	37.48	90.46
6afa^b	26.37	27.12	27.00	31.63	28.99	29.66	29.63	44.16	48.45
6pub	6.66	12.40	10.53	8.34	6.80	9.61	8.60	7.33	28.95
6v83	52.56	65.47	64.35	60.22	48.53	65.99	63.45	57.82	66.62
5l7i^c	79.21	79.59	81.24	89.54	79.04	80.36	80.46	89.39	21.39
Average	23.65	26.70	25.36	26.96	24.47	27.74	26.10	27.97	55.05

a. The strain energy contributed from the bonded (70 %) and nonbonded (30 %) forms. b. The strain energy contributed from the bonded (50 %) and nonbonded (50 %) forms. c. Total strain energy of drugs/inhibitors in chain A and chain B.

Supplementary Table 16 Bond RMSD (Å) of the all 50 drugs/inhibitors in the proteins from our quantum refinement calculations (M1-M10) with respect to M7 as the reference. Those results for X-ray were taken from the experimental structure without our further refinement.

PDB ID	M1	M3	M5	M6	M8	M10	M6R	X-ray
1xbb	0.006	0.005	0.005	0.008	0.002	0.004	0.007	0.020
2w26	0.008	0.010	0.009	0.012	0.008	0.008	0.011	0.037
4hzz	0.014	0.009	0.025	0.019	0.012	0.007	0.007	0.059
4kra	0.017	0.017	0.016	0.020	0.008	0.007	0.015	0.025
4p6w	0.003	0.007	0.007	0.008	0.006	0.006	0.007	0.042
5hls	0.004	0.006	0.008	0.012	0.006	0.008	0.010	0.022
5kr1	0.004	0.006	0.005	0.010	0.004	0.005	0.006	0.075
5p9i	0.005	0.005	0.006	0.009	0.003	0.004	0.015	0.015
6jx4	0.006	0.007	0.008	0.012	0.008	0.006	0.007	0.052
7rfw	0.007	0.008	0.008	0.010	0.008	0.009	0.011	0.030
(bonded)								
7rfw	0.007	0.008	0.010	0.009	0.004	0.004	0.007	0.030
(non-bonded)								
3qqa	0.003	0.006	0.007	0.010	0.006	0.006	0.009	0.046
4xh6	0.007	0.008	0.006	0.009	0.006	0.005	0.005	0.023
4yhm	0.009	0.016	0.009	0.021	0.011	0.005	0.010	0.024
6v83	0.012	0.018	0.016	0.021	0.014	0.011	0.016	0.030
1ajx	0.005	0.005	0.008	0.008	0.004	0.004	0.007	0.025
1br5	0.011	0.013	0.011	0.012	0.006	0.006	0.009	0.046
1d4i	0.006	0.007	0.007	0.010	0.004	0.004	0.006	0.023
1f9g	0.004	0.023	0.010	0.010	0.023	0.010	0.009	0.039
1lf3	0.003	0.004	0.005	0.008	0.003	0.004	0.007	0.038
4zb8	0.012	0.021	0.017	0.025	0.014	0.007	0.012	0.044
1p1o	0.020	0.011	0.027	0.039	0.011	0.005	0.013	0.018
1pf7	0.007	0.013	0.011	0.012	0.011	0.008	0.007	0.055
1m7q	0.005	0.007	0.006	0.009	0.006	0.005	0.008	0.060
1pa9	0.021	0.027	0.019	0.025	0.022	0.012	0.012	0.058
1xoz	0.002	0.004	0.003	0.005	0.003	0.003	0.005	0.022
2ax6	0.010	0.011	0.010	0.011	0.006	0.006	0.008	0.066
2iwu	0.006	0.017	0.013	0.010	0.015	0.011	0.008	0.058
2p7z	0.007	0.008	0.008	0.009	0.009	0.004	0.010	0.033
1uy9	0.003	0.002	0.005	0.008	0.004	0.005	0.011	0.018
3eq7	0.006	0.007	0.007	0.010	0.003	0.003	0.006	0.037
5y1y	0.010	0.021	0.010	0.013	0.017	0.005	0.008	0.053
5y62	0.006	0.010	0.011	0.014	0.010	0.011	0.011	0.056
1nox	0.004	0.011	0.009	0.011	0.008	0.007	0.010	0.041
1nj1	0.011	0.018	0.019	0.017	0.017	0.016	0.012	0.029
2a3i	0.003	0.005	0.006	0.005	0.004	0.004	0.007	0.053
5ct1	0.009	0.013	0.011	0.022	0.014	0.007	0.013	0.054

6afa	0.018	0.023	0.023	0.014	0.007	0.007	0.025	0.043
(bonded)								
6afa	0.011	0.022	0.023	0.020	0.015	0.016	0.038	0.180
(non-bonded)								
6pub	0.003	0.012	0.007	0.008	0.012	0.005	0.008	0.034
1owk	0.010	0.021	0.009	0.008	0.014	0.004	0.008	0.030
2qbr	0.011	0.018	0.019	0.020	0.017	0.008	0.008	0.038
1yat	0.003	0.006	0.006	0.006	0.004	0.004	0.006	0.025
1xp9	0.006	0.011	0.011	0.010	0.012	0.008	0.009	0.045
2hxm	0.007	0.010	0.009	0.017	0.009	0.005	0.012	0.042
2wca	0.008	0.010	0.010	0.011	0.004	0.004	0.009	0.025
2wi4	0.006	0.007	0.009	0.014	0.005	0.006	0.015	0.031
1uyg	0.006	0.005	0.005	0.008	0.005	0.004	0.008	0.066
2wi1	0.005	0.004	0.005	0.006	0.003	0.004	0.005	0.013
3fr2	0.004	0.005	0.005	0.008	0.006	0.006	0.007	0.040
3atm	0.008	0.005	0.013	0.008	0.007	0.004	0.009	0.026
5l7i	0.006	0.006	0.006	0.013	0.004	0.005	0.011	0.022

Supplementary Table 17 Bond MAD (Å) of the all 50 drugs/inhibitors in the proteins from our quantum refinement calculations (M1-M10) with respect to M7 as the reference. Those results for X-ray were taken from the experimental structure without our further refinement.

PDB ID	M1	M3	M5	M6	M8	M10	M6R	X-ray
1xbb	0.006	0.005	0.005	0.008	0.002	0.004	0.007	0.020
2w26	0.008	0.010	0.009	0.012	0.008	0.008	0.011	0.037
4hzz	0.014	0.009	0.025	0.019	0.012	0.007	0.007	0.059
4kra	0.017	0.017	0.016	0.020	0.008	0.007	0.015	0.025
4p6w	0.003	0.007	0.007	0.008	0.006	0.006	0.007	0.042
5hls	0.004	0.006	0.008	0.012	0.006	0.008	0.010	0.022
5kr1	0.004	0.006	0.005	0.010	0.004	0.005	0.006	0.075
5p9i	0.005	0.005	0.006	0.009	0.003	0.004	0.015	0.015
6jx4	0.006	0.007	0.008	0.012	0.008	0.006	0.007	0.052
7rfw	0.007	0.008	0.008	0.010	0.008	0.009	0.011	0.030
(bonded)								
7rfw	0.007	0.008	0.010	0.009	0.004	0.004	0.007	0.030
(non-bonded)								
3qqa	0.003	0.006	0.007	0.010	0.006	0.006	0.009	0.046
4xh6	0.007	0.008	0.006	0.009	0.006	0.005	0.005	0.023
4yhm	0.009	0.016	0.009	0.021	0.011	0.005	0.010	0.024
6v83	0.012	0.018	0.016	0.021	0.014	0.011	0.016	0.030
1ajx	0.005	0.005	0.008	0.008	0.004	0.004	0.007	0.025
1br5	0.011	0.013	0.011	0.012	0.006	0.006	0.009	0.046
1d4i	0.006	0.007	0.007	0.010	0.004	0.004	0.006	0.023
1f9g	0.004	0.023	0.010	0.010	0.023	0.010	0.009	0.039
1lf3	0.003	0.004	0.005	0.008	0.003	0.004	0.007	0.038
4zb8	0.012	0.021	0.017	0.025	0.014	0.007	0.012	0.044
1p1o	0.020	0.011	0.027	0.039	0.011	0.005	0.013	0.018
1pf7	0.007	0.013	0.011	0.012	0.011	0.008	0.007	0.055
1m7q	0.005	0.007	0.006	0.009	0.006	0.005	0.008	0.060
1pa9	0.021	0.027	0.019	0.025	0.022	0.012	0.012	0.058
1xoz	0.002	0.004	0.003	0.005	0.003	0.003	0.005	0.022
2ax6	0.010	0.011	0.010	0.011	0.006	0.006	0.008	0.066
2iwu	0.006	0.017	0.013	0.010	0.015	0.011	0.008	0.058
2p7z	0.007	0.008	0.008	0.009	0.009	0.004	0.010	0.033
1uy9	0.003	0.002	0.005	0.008	0.004	0.005	0.011	0.018
3eq7	0.006	0.007	0.007	0.010	0.003	0.003	0.006	0.037
5y1y	0.010	0.021	0.010	0.013	0.017	0.005	0.008	0.053
5y62	0.006	0.010	0.011	0.014	0.010	0.011	0.011	0.056
1nox	0.004	0.011	0.009	0.011	0.008	0.007	0.010	0.041
1nj1	0.011	0.018	0.019	0.017	0.017	0.016	0.012	0.029
2a3i	0.003	0.005	0.006	0.005	0.004	0.004	0.007	0.053
5ct1	0.009	0.013	0.011	0.022	0.014	0.007	0.013	0.054

6afa	0.018	0.023	0.023	0.014	0.007	0.007	0.025	0.043
(bonded)								
6afa	0.011	0.022	0.023	0.020	0.015	0.016	0.038	0.180
(non-bonded)								
6pub	0.003	0.012	0.007	0.008	0.012	0.005	0.008	0.034
1owk	0.010	0.021	0.009	0.008	0.014	0.004	0.008	0.030
2qbr	0.011	0.018	0.019	0.020	0.017	0.008	0.008	0.038
1yat	0.003	0.006	0.006	0.006	0.004	0.004	0.006	0.025
1xp9	0.006	0.011	0.011	0.010	0.012	0.008	0.009	0.045
2hxm	0.007	0.010	0.009	0.017	0.009	0.005	0.012	0.042
2wca	0.008	0.010	0.010	0.011	0.004	0.004	0.009	0.025
2wi4	0.006	0.007	0.009	0.014	0.005	0.006	0.015	0.031
1uyg	0.006	0.005	0.005	0.008	0.005	0.004	0.008	0.066
2wi1	0.005	0.004	0.005	0.006	0.003	0.004	0.005	0.013
3fr2	0.004	0.005	0.005	0.008	0.006	0.006	0.007	0.040
3atm	0.008	0.005	0.013	0.008	0.007	0.004	0.009	0.026
5l7i	0.006	0.006	0.006	0.013	0.004	0.005	0.011	0.022

Supplementary Table 18 Bond R^2 of the all 50 drugs/inhibitors in the proteins from our quantum refinement calculations (**M1-M10**) with respect to **M7** as the reference. Those results for X-ray were taken from the experimental structure without our further refinement.

PDB ID	M1	M3	M5	M6	M8	M10	M6R	X-ray
1xbb	0.991	0.993	0.993	0.982	0.999	0.996	0.988	0.894
2w26	0.996	0.994	0.995	0.991	0.996	0.996	0.993	0.911
4hzz	0.982	0.991	0.936	0.963	0.985	0.996	0.995	0.647
4kra	0.957	0.957	0.960	0.942	0.990	0.993	0.965	0.905
4p6w	0.999	0.997	0.997	0.996	0.998	0.998	0.997	0.892
5hls	0.998	0.996	0.992	0.982	0.996	0.993	0.987	0.941
5kr1	0.998	0.996	0.997	0.989	0.998	0.998	0.996	0.376
5p9i	0.996	0.995	0.993	0.984	0.998	0.996	0.955	0.955
6jx4	0.990	0.985	0.980	0.958	0.980	0.990	0.984	0.219
7rfw								
(bonded)	0.997	0.997	0.997	0.994	0.996	0.996	0.993	0.953
7rfw								
(non-bonded)	0.997	0.996	0.994	0.995	0.999	0.999	0.996	0.940
3qqa	0.999	0.996	0.993	0.986	0.996	0.995	0.989	0.726
4xh6	0.974	0.966	0.979	0.953	0.982	0.987	0.988	0.719
4yhm	0.986	0.956	0.986	0.924	0.978	0.995	0.984	0.903
6v83	0.995	0.987	0.989	0.982	0.992	0.995	0.990	0.965
1ajx	0.993	0.994	0.985	0.980	0.995	0.995	0.987	0.825
1br5	0.983	0.976	0.982	0.980	0.995	0.995	0.989	0.690
1d4i	0.991	0.990	0.989	0.981	0.997	0.996	0.993	0.890
1f9g	0.998	0.947	0.989	0.989	0.945	0.989	0.991	0.837
1lf3	0.999	0.996	0.996	0.988	0.998	0.997	0.989	0.712
4zb8	0.996	0.989	0.993	0.985	0.995	0.999	0.996	0.951
1p1o	0.968	0.990	0.943	0.883	0.991	0.998	0.987	0.976
1pf7	0.994	0.982	0.987	0.984	0.985	0.994	0.994	0.657
1m7q	0.997	0.994	0.996	0.990	0.996	0.997	0.993	0.610
1pa9	0.963	0.935	0.967	0.947	0.959	0.987	0.987	0.709
1xoz	0.999	0.997	0.998	0.995	0.998	0.999	0.994	0.907
2ax6	0.989	0.984	0.987	0.985	0.995	0.996	0.992	0.472
2iwu	0.998	0.981	0.988	0.994	0.984	0.991	0.995	0.766
2p7z	0.982	0.980	0.977	0.972	0.973	0.994	0.966	0.637
1uy9	0.998	0.999	0.993	0.984	0.996	0.994	0.973	0.926
3eq7	0.996	0.994	0.994	0.990	0.999	0.999	0.995	0.844
5y1y	0.978	0.906	0.981	0.964	0.938	0.994	0.987	0.395
5y62	0.996	0.991	0.989	0.983	0.992	0.988	0.989	0.726
1nox	0.998	0.987	0.991	0.987	0.992	0.995	0.989	0.813
1nj1	0.990	0.975	0.972	0.979	0.977	0.980	0.989	0.937
2a3i	0.999	0.997	0.996	0.998	0.998	0.998	0.995	0.673
5ct1	0.989	0.980	0.986	0.941	0.977	0.993	0.980	0.648

6afa								
(bonded)	0.989	0.983	0.982	0.993	0.998	0.998	0.979	0.937
6afa								
(non-bonded)	0.999	0.996	0.995	0.996	0.998	0.998	0.986	0.702
6pub	0.992	0.856	0.956	0.940	0.866	0.973	0.942	-0.061
1owk	0.983	0.921	0.984	0.987	0.961	0.997	0.988	0.832
2qbr	0.993	0.983	0.981	0.980	0.985	0.997	0.997	0.925
1yat	0.999	0.996	0.996	0.996	0.998	0.998	0.996	0.924
1xp9	0.997	0.988	0.990	0.991	0.987	0.995	0.993	0.815
2hxm	0.992	0.984	0.986	0.958	0.988	0.996	0.979	0.730
2wca	0.991	0.986	0.987	0.983	0.998	0.997	0.989	0.908
2wi4	0.997	0.995	0.993	0.983	0.998	0.996	0.981	0.911
1uyg	0.983	0.988	0.988	0.970	0.989	0.993	0.973	-1.052
2wi1	0.995	0.997	0.992	0.992	0.998	0.995	0.993	0.952
3fr2	0.997	0.995	0.995	0.986	0.993	0.993	0.991	0.667
3atm	0.976	0.991	0.932	0.975	0.983	0.993	0.967	0.726
5l7i	0.998	0.998	0.998	0.991	0.999	0.999	0.994	0.973

Supplementary Table 19 Angle RMSD (°) of the all 50 drugs/inhibitors in the proteins from our quantum refinement calculations (M1-M10) with respect to M7 as the reference. Those results for X-ray were taken from the experimental structure without our further refinement.

PDB ID	M1	M3	M5	M6	M8	M10	M6R	X-ray
1xbb	1.1	0.8	1.1	1.2	0.5	1.0	1.5	3.4
2w26	0.6	0.8	0.8	0.8	0.6	0.7	0.7	3.2
4hzz	1.0	2.5	1.0	1.1	2.8	0.6	0.8	5.0
4kra	0.9	1.6	1.1	0.9	1.2	0.7	0.9	5.0
4p6w	0.4	0.7	0.5	1.0	0.6	0.4	0.8	3.6
5hls	0.4	0.9	0.7	0.6	0.9	0.8	0.6	1.9
5kr1	0.5	0.8	0.7	1.0	0.4	0.4	1.0	3.0
5p9i	0.3	0.5	0.5	0.8	0.3	0.3	2.1	2.1
6jx4	0.9	1.0	0.8	0.9	0.9	0.6	0.4	2.2
7rfw								
(bonded)	0.7	0.8	0.9	1.0	0.7	0.6	0.8	3.4
7rfw								
(non-bonded)	1.3	1.3	1.3	1.2	0.6	0.5	1.0	6.2
3qqa	0.2	0.6	0.7	0.5	0.6	0.5	0.5	5.3
4xh6	0.3	0.6	0.5	0.9	0.5	0.4	0.7	1.9
4yhm	0.8	1.1	0.7	1.4	1.2	0.4	0.7	3.1
6v83	1.1	2.1	1.8	1.6	2.1	1.7	1.1	3.0
1ajx	0.4	0.4	0.6	0.5	0.4	0.4	0.4	3.1
1br5	1.0	1.2	1.0	1.1	0.7	0.7	1.0	2.9
1d4i	0.3	0.3	0.4	0.5	0.3	0.3	0.6	2.1
1f9g	0.4	1.8	1.0	0.7	1.6	0.7	0.6	4.0
1lf3	0.3	0.5	0.5	0.6	0.4	0.4	0.6	3.6
4zb8	0.8	1.5	0.7	1.0	1.6	0.7	1.0	2.4
1p1o	1.6	1.6	1.8	2.4	2.0	0.5	1.0	3.1
1pf7	0.9	1.2	1.6	1.2	0.8	0.5	0.8	2.8
1m7q	0.3	0.5	0.4	0.6	0.5	0.4	0.5	3.2
1pa9	0.9	1.4	1.1	1.3	1.5	1.1	0.9	5.2
1xoz	0.2	0.4	0.3	0.4	0.3	0.3	0.4	2.7
2ax6	0.4	0.5	0.5	0.5	0.4	0.3	0.8	1.8
2iwu	0.7	1.1	1.0	0.9	1.0	0.9	0.8	4.5
2p7z	0.8	1.4	0.9	1.1	1.3	0.7	0.7	1.9
1uy9	0.4	0.3	0.5	0.8	0.4	0.4	0.9	3.0
3eq7	0.3	0.4	0.4	0.7	0.3	0.3	0.5	2.3
5y1y	0.7	0.9	0.6	0.7	1.0	0.5	0.4	2.6
5y62	0.5	0.6	0.6	1.8	0.5	0.6	0.8	5.2
1nox	0.5	0.8	0.6	0.6	0.6	0.5	0.6	5.6
1nj1	1.1	1.5	1.3	1.3	1.3	1.3	1.1	3.7
2a3i	0.5	0.6	0.5	0.6	0.4	0.4	0.6	3.7
5ct1	0.7	0.9	1.4	0.8	1.2	1.1	0.4	3.4

6afa								
(bonded)	0.8	1.3	1.4	1.1	1.0	1.0	2.0	4.9
6afa								
(non-bonded)	1.5	1.6	1.6	1.5	0.9	0.9	1.1	6.7
6pub	0.5	0.6	0.7	0.7	0.5	0.4	0.4	1.6
1owk	0.7	1.4	0.8	1.1	0.9	0.3	0.7	1.6
2qbr	1.1	1.2	1.2	1.2	2.3	0.7	0.6	2.3
1yat	0.2	0.5	0.4	0.5	0.4	0.3	0.5	2.6
1xp9	0.3	0.6	0.6	0.6	0.7	0.7	0.5	3.4
2hxm	0.8	1.9	0.9	1.2	1.7	0.6	1.0	8.6
2wca	0.4	0.6	0.5	1.0	0.4	0.3	0.8	2.4
2wi4	0.3	0.5	0.5	1.0	0.4	0.4	0.8	2.8
1uyg	0.5	0.5	0.6	0.6	0.5	0.5	0.6	2.3
2wi1	1.0	0.8	0.8	0.6	0.4	0.3	0.7	4.1
3fr2	0.6	1.1	0.6	0.7	1.1	0.5	0.6	3.4
3atm	0.3	0.5	0.4	0.3	0.5	0.2	0.3	2.1
5l7i	0.4	0.5	0.5	0.8	0.4	0.3	0.8	4.0

Supplementary Table 20 Angle MAD (°) of the all 50 drugs/inhibitors in the proteins from our quantum refinement calculations (M1-M10) with respect to M7 as the reference. Those results for X-ray were taken from the experimental structure without our further refinement.

PDB ID	M1	M3	M5	M6	M8	M10	M6R	X-ray
1xbb	0.8	0.6	0.7	0.9	0.3	0.6	1.0	2.6
2w26	0.4	0.6	0.6	0.6	0.5	0.5	0.6	2.5
4hzz	0.7	1.8	0.7	0.9	1.8	0.4	0.6	4.0
4kra	0.7	1.3	0.9	0.7	0.9	0.6	0.7	3.8
4p6w	0.3	0.6	0.5	0.8	0.5	0.3	0.6	2.6
5hls	0.3	0.6	0.5	0.5	0.6	0.6	0.5	1.6
5kr1	0.4	0.5	0.5	0.7	0.3	0.3	0.7	2.2
5p9i	0.3	0.3	0.4	0.6	0.3	0.3	1.6	1.6
6jx4	0.5	0.7	0.6	0.6	0.5	0.4	0.3	1.7
7rfw								
(bonded)	0.5	0.6	0.7	0.7	0.5	0.5	0.7	2.1
7rfw								
(non-bonded)	0.7	0.7	0.7	0.9	0.5	0.4	0.7	2.6
3qqa	0.2	0.5	0.6	0.4	0.4	0.4	0.4	4.1
4xh6	0.2	0.5	0.5	0.7	0.4	0.3	0.5	1.5
4yhm	0.6	0.9	0.5	0.9	0.7	0.2	0.5	2.1
6v83	0.8	1.4	1.4	1.2	1.3	1.2	0.9	2.2
1ajx	0.3	0.3	0.4	0.4	0.3	0.3	0.3	2.2
1br5	0.8	1.0	0.8	0.9	0.5	0.5	0.8	2.5
1d4i	0.2	0.2	0.3	0.3	0.2	0.2	0.4	1.7
1f9g	0.4	1.5	0.8	0.6	1.3	0.6	0.5	3.0
1lf3	0.2	0.3	0.3	0.5	0.3	0.3	0.4	2.5
4zb8	0.6	1.2	0.5	0.7	1.2	0.5	0.8	1.9
1p1o	1.1	1.1	1.2	1.5	1.3	0.4	0.8	2.4
1pf7	0.7	1.0	1.1	0.9	0.6	0.4	0.7	2.2
1m7q	0.2	0.4	0.3	0.5	0.3	0.3	0.4	2.2
1pa9	0.6	1.1	0.9	1.0	1.2	0.9	0.8	4.1
1xoz	0.2	0.3	0.3	0.4	0.3	0.2	0.3	2.3
2ax6	0.3	0.4	0.4	0.4	0.3	0.2	0.7	1.3
2iwu	0.5	0.9	0.8	0.7	0.8	0.7	0.7	3.8
2p7z	0.6	1.0	0.6	0.8	0.9	0.4	0.5	1.4
1uy9	0.3	0.2	0.4	0.7	0.3	0.3	0.7	2.3
3eq7	0.2	0.3	0.3	0.5	0.2	0.2	0.4	1.8
5y1y	0.5	0.8	0.5	0.6	0.8	0.4	0.4	2.2
5y62	0.4	0.5	0.5	1.2	0.4	0.4	0.6	3.8
1nox	0.4	0.6	0.5	0.5	0.5	0.4	0.5	4.2
1nj1	0.8	1.1	1.0	1.1	0.9	0.9	0.9	2.8
2a3i	0.4	0.5	0.4	0.5	0.3	0.3	0.5	2.7
5ct1	0.5	0.8	1.1	0.7	1.0	0.8	0.4	2.7

6afa								
(bonded)	0.6	0.9	1.0	0.8	0.7	0.7	1.2	3.8
6afa								
(non-bonded)	1.1	1.3	1.3	1.1	0.5	0.6	0.8	4.7
6pub	0.3	0.5	0.6	0.6	0.3	0.3	0.4	1.3
1owk	0.5	1.0	0.6	0.9	0.7	0.3	0.6	1.4
2qbr	0.7	0.9	0.8	0.8	1.5	0.5	0.5	1.7
1yat	0.2	0.4	0.3	0.4	0.3	0.3	0.4	2.0
1xp9	0.3	0.5	0.5	0.5	0.6	0.5	0.4	2.6
2hxm	0.6	1.4	0.7	0.9	1.2	0.5	0.7	4.6
2wca	0.3	0.4	0.4	0.7	0.3	0.3	0.6	1.7
2wi4	0.2	0.4	0.4	0.8	0.3	0.3	0.6	2.2
1uyg	0.4	0.4	0.4	0.5	0.4	0.4	0.5	1.9
2wi1	0.8	0.6	0.7	0.5	0.4	0.3	0.6	3.3
3fr2	0.4	0.8	0.4	0.5	0.8	0.3	0.5	2.4
3atm	0.3	0.3	0.3	0.2	0.4	0.1	0.3	1.7
5l7i	0.3	0.4	0.4	0.6	0.3	0.2	0.6	2.8

Supplementary Table 21 Angle R^2 of the all 50 drugs/inhibitors in the proteins from our quantum refinement calculations (**M1-M10**) with respect to **M7** as the reference. Those results for X-ray were taken from the experimental structure without our further refinement.

PDB ID	M1	M3	M5	M6	M8	M10	M6R	X-ray
1xbb	0.924	0.966	0.924	0.916	0.982	0.945	0.875	0.324
2w26	0.992	0.987	0.986	0.987	0.992	0.989	0.989	0.797
4hzz	0.969	0.796	0.969	0.961	0.733	0.990	0.980	0.175
4kra	0.966	0.897	0.949	0.963	0.942	0.977	0.966	-0.032
4p6w	0.997	0.989	0.994	0.978	0.992	0.996	0.984	0.706
5hls	0.997	0.979	0.988	0.991	0.980	0.985	0.991	0.910
5kr1	0.994	0.987	0.990	0.979	0.997	0.997	0.978	0.796
5p9i	0.997	0.995	0.995	0.987	0.997	0.997	0.892	0.892
6jx4	0.982	0.975	0.985	0.982	0.983	0.993	0.997	0.879
7rfw								
(bonded)	0.988	0.984	0.980	0.975	0.990	0.991	0.984	0.740
7rfw								
(non-bonded)	0.978	0.978	0.978	0.982	0.995	0.997	0.987	0.499
3qqa	0.999	0.992	0.988	0.994	0.992	0.993	0.994	0.344
4xh6	0.982	0.937	0.951	0.866	0.960	0.973	0.928	0.389
4yhm	0.980	0.961	0.985	0.936	0.955	0.996	0.985	0.708
6v83	0.979	0.927	0.948	0.955	0.927	0.952	0.980	0.851
1ajx	0.990	0.990	0.978	0.988	0.989	0.992	0.992	0.471
1br5	0.953	0.935	0.957	0.944	0.980	0.981	0.952	0.623
1d4i	0.998	0.998	0.997	0.996	0.998	0.999	0.993	0.905
1f9g	0.998	0.959	0.987	0.993	0.968	0.993	0.995	0.798
1lf3	0.998	0.994	0.995	0.990	0.997	0.996	0.992	0.664
4zb8	0.987	0.950	0.990	0.979	0.945	0.990	0.976	0.873
1p1o	0.961	0.963	0.953	0.912	0.941	0.996	0.986	0.858
1pf7	0.989	0.978	0.964	0.978	0.991	0.996	0.992	0.890
1m7q	0.995	0.980	0.991	0.975	0.984	0.988	0.984	0.322
1pa9	0.982	0.957	0.974	0.962	0.945	0.972	0.980	0.365
1xoz	0.999	0.997	0.998	0.996	0.998	0.999	0.997	0.860
2ax6	0.995	0.992	0.993	0.993	0.997	0.998	0.982	0.912
2iwu	0.977	0.944	0.954	0.959	0.947	0.955	0.965	0.002
2p7z	0.954	0.861	0.947	0.906	0.878	0.965	0.967	0.739
1uy9	0.997	0.999	0.996	0.988	0.998	0.998	0.986	0.850
3eq7	0.999	0.997	0.998	0.993	0.998	0.998	0.997	0.931
5y1y	0.960	0.929	0.969	0.963	0.918	0.975	0.984	0.408
5y62	0.996	0.995	0.996	0.959	0.997	0.995	0.992	0.646
1nox	0.993	0.980	0.989	0.987	0.990	0.993	0.990	0.036
1nj1	0.983	0.966	0.974	0.975	0.973	0.974	0.983	0.794
2a3i	0.992	0.988	0.992	0.988	0.994	0.994	0.989	0.551
5ct1	0.978	0.959	0.901	0.967	0.927	0.940	0.990	0.444

6afa								
(bonded)	0.991	0.974	0.971	0.984	0.986	0.986	0.944	0.641
6afa								
(non-bonded)	0.984	0.983	0.983	0.984	0.995	0.994	0.992	0.699
6pub	0.953	0.916	0.898	0.897	0.954	0.964	0.956	0.445
1owk	0.981	0.924	0.977	0.949	0.965	0.996	0.981	0.896
2qbr	0.969	0.963	0.963	0.964	0.862	0.987	0.990	0.873
1yat	0.999	0.992	0.994	0.993	0.994	0.996	0.993	0.790
1xp9	0.997	0.988	0.988	0.990	0.985	0.988	0.994	0.671
2hxm	0.968	0.829	0.965	0.931	0.866	0.982	0.952	-2.545
2wca	0.994	0.990	0.993	0.971	0.996	0.997	0.982	0.817
2wi4	0.988	0.970	0.969	0.865	0.978	0.979	0.912	-0.074
1uyg	0.993	0.993	0.993	0.992	0.994	0.993	0.991	0.877
2wi1	0.896	0.946	0.933	0.963	0.982	0.989	0.955	-0.630
3fr2	0.993	0.974	0.991	0.990	0.974	0.996	0.991	0.764
3atm	0.998	0.996	0.998	0.999	0.996	1.000	0.998	0.937
5l7i	0.996	0.994	0.994	0.983	0.997	0.998	0.985	0.621

Supplementary Table 22 Dihedral RMSD (°) of the all 50 drugs/inhibitors in the proteins from our quantum refinement calculations (M1-M10) with respect to M7 as the reference. Those results for X-ray were taken from the experimental structure without our further refinement.

PDB ID	M1	M3	M5	M6	M8	M10	M6R	X-ray
1xbb	1.5	2.6	3.5	4.1	1.8	3.2	3.9	18.3
2w26	3.3	2.3	1.9	1.1	1.5	2.9	2.3	7.8
4hzz	1.0	2.1	1.2	1.5	2.1	0.7	0.9	8.8
4kra	3.8	4.7	3.6	3.3	3.5	1.4	2.7	13.3
4p6w	1.3	2.7	2.3	1.7	1.8	1.5	1.2	6.1
5hls	0.8	2.5	2.6	0.8	2.6	2.4	0.6	4.3
5kr1	0.8	1.2	1.0	1.2	0.7	0.7	2.4	3.2
5p9i	0.7	0.7	1.3	0.9	0.5	0.6	3.0	3.0
6jx4	2.0	4.2	1.9	2.3	3.0	1.5	0.5	9.9
7rfw	1.8	1.5	2.0	1.7	0.9	1.3	1.6	8.5
(bonded)								
7rfw	8.0	8.0	7.9	6.0	1.3	1.1	2.4	7.9
(non-bonded)								
3qqa	0.6	0.7	1.4	0.7	0.9	0.9	0.5	5.4
4xh6	1.0	0.7	0.4	1.5	0.8	0.6	1.5	2.6
4yhm	1.0	1.0	1.7	1.6	1.6	0.7	2.1	8.5
6v83	3.8	5.2	5.3	3.8	5.4	5.1	1.6	8.9
1ajx	0.9	1.1	1.2	0.9	0.8	0.7	0.5	12.6
1br5	2.6	2.7	2.4	4.1	1.9	1.9	2.7	4.3
1d4i	0.5	0.7	0.6	0.7	0.6	0.5	0.8	5.0
1f9g	0.6	1.6	1.3	1.0	1.3	0.8	0.9	9.8
1lf3	0.8	1.5	1.1	1.5	1.0	1.0	1.6	13.5
4zb8	0.9	1.3	1.5	1.3	1.2	1.0	1.0	4.7
1p1o	2.9	2.6	3.2	2.9	1.7	0.9	1.7	3.7
1pf7	1.4	1.0	6.3	0.9	1.1	0.8	1.0	7.6
1m7q	0.6	1.8	0.9	1.5	1.1	1.3	1.0	6.7
1pa9	3.2	3.4	3.4	3.1	1.2	1.3	1.1	3.4
1xoz	0.2	0.7	0.4	0.7	0.8	0.5	0.7	2.2
2ax6	0.6	0.8	0.7	0.7	0.4	0.3	0.4	11.4
2p7z	4.0	4.4	3.3	4.3	4.0	1.7	1.4	9.6
1uy9	0.5	0.5	0.5	0.9	0.4	0.4	0.7	1.3
3eq7	0.7	1.0	1.1	1.1	0.6	0.6	0.6	6.9
5y1y	1.1	1.0	3.2	0.5	1.3	0.5	0.3	3.6
5y62	0.7	0.9	1.3	10.3	1.1	2.6	1.0	7.3
1nox	0.7	1.4	1.2	1.3	1.1	0.8	0.7	6.1
1nj1	1.4	1.9	1.8	1.6	1.8	1.7	1.7	9.9
2a3i	1.4	1.6	1.2	1.4	1.1	0.9	0.7	7.0
5ct1	0.3	0.7	2.4	0.7	0.8	0.3	0.6	1.2
6afa	0.8	1.0	0.9	1.8	0.7	0.6	1.5	4.6

(bonded)								
6afa	1.7	1.9	1.9	2.5	0.3	0.3	1.0	3.0
(non-bonded)								
6pub	1.3	7.7	5.9	3.3	1.8	1.2	0.7	19.8
1owk	1.1	1.7	1.4	1.2	1.3	0.8	1.0	5.5
2qbr	1.5	2.3	2.4	2.1	3.7	1.6	1.2	6.0
1yat	0.3	0.6	0.5	0.7	0.6	0.5	0.7	6.4
1xp9	0.9	0.8	0.8	1.3	1.0	0.7	0.8	5.9
2hxm	1.5	1.8	1.4	1.7	1.3	0.7	0.8	6.8
2wca	0.6	0.8	1.0	0.8	0.5	0.5	0.5	8.9
2wi4	0.9	1.2	0.8	0.7	1.8	0.8	0.5	16.3
1uyg	3.2	3.2	2.9	2.7	1.0	2.3	3.4	6.9
2wi1	4.0	1.6	1.7	0.6	1.1	1.0	0.5	12.5
3fr2	0.9	1.4	1.0	0.8	1.8	0.9	0.9	4.6
3atm	0.3	0.5	0.4	0.3	1.0	0.2	0.6	2.5
5l7i	0.4	0.7	1.0	1.4	0.8	0.8	1.6	25.2

Supplementary Table 23 Dihedral MAD ($^{\circ}$) of the all 50 drugs/inhibitors in the proteins from our quantum refinement calculations (M1-M10) with respect to M7 as the reference. Those results for X-ray were taken from the experimental structure without our further refinement.

PDB ID	M1	M3	M5	M6	M8	M10	M6R	X-ray
1xbb	1.2	1.9	2.3	3.1	1.4	2.3	2.6	15.2
2w26	2.6	1.8	1.5	0.9	1.1	2.2	2.0	5.9
4hzz	0.8	1.7	1.0	1.3	1.6	0.6	0.8	6.8
4kra	2.9	2.9	2.6	2.6	2.5	1.1	2.3	9.2
4p6w	0.9	1.9	1.7	1.3	1.3	1.3	1.0	5.1
5hls	0.6	2.3	2.1	0.7	2.3	2.0	0.5	3.5
5kr1	0.7	1.0	0.8	1.0	0.5	0.5	1.6	2.6
5p9i	0.6	0.6	1.0	0.7	0.4	0.5	1.9	1.9
6jx4	1.7	3.2	1.5	1.9	2.2	1.1	0.4	7.0
7rfw	1.3	1.3	1.6	1.4	0.7	1.0	1.2	5.2
(bonded)								
7rfw	3.2	3.2	2.9	3.0	1.0	0.9	1.9	5.1
(non-bonded)								
3qqa	0.5	0.6	1.3	0.5	0.7	0.7	0.4	4.5
4xh6	0.9	0.7	0.3	1.3	0.6	0.5	0.9	2.5
4yhm	0.8	0.8	1.4	1.2	1.3	0.5	1.9	6.0
6v83	2.6	3.7	4.0	2.8	3.6	4.0	1.3	6.7
1ajx	0.8	0.8	1.0	0.7	0.6	0.6	0.4	10.3
1br5	1.9	1.9	1.8	3.2	1.3	1.5	2.2	3.4
1d4i	0.4	0.6	0.5	0.5	0.5	0.4	0.6	4.0
1f9g	0.5	1.3	1.1	0.9	1.1	0.8	0.8	8.0
1lf3	0.6	1.2	0.9	1.2	0.8	0.8	1.3	9.3
4zb8	0.7	1.0	1.2	0.9	0.9	0.8	0.8	3.7
1p1o	2.2	1.8	2.4	2.1	1.3	0.7	1.3	2.9
1pf7	1.2	0.8	5.0	0.8	1.0	0.7	0.8	6.6
1m7q	0.4	1.5	0.8	1.1	0.9	1.1	0.8	5.2
1pa9	2.9	3.2	3.3	2.6	0.9	1.1	0.9	2.9
1xoz	0.1	0.6	0.3	0.6	0.7	0.4	0.7	1.8
2ax6	0.4	0.7	0.6	0.5	0.4	0.3	0.4	7.0
2p7z	3.2	3.5	2.4	3.4	3.1	1.1	1.0	7.8
1uy9	0.4	0.4	0.4	0.7	0.3	0.3	0.6	1.1
3eq7	0.6	0.9	0.9	0.9	0.5	0.5	0.5	6.4
5y1y	1.0	1.0	2.9	0.4	1.2	0.5	0.2	3.0
5y62	0.6	0.8	1.2	9.2	0.9	2.3	0.9	6.1
1nox	0.6	1.1	0.9	1.0	0.9	0.7	0.6	4.5
1nj1	1.2	1.5	1.3	1.2	1.4	1.4	1.4	7.3
2a3i	1.1	1.2	1.0	1.2	0.8	0.9	0.5	5.1
5ct1	0.2	0.7	2.4	0.5	0.6	0.3	0.5	1.0
6afa	0.6	0.8	0.8	1.7	0.6	0.5	1.3	3.9

(bonded)								
6afa	1.6	1.7	1.7	2.2	0.3	0.3	0.8	2.3
(non-bonded)								
6pub	1.0	6.8	5.0	2.7	1.6	0.9	0.6	17.6
1owk	0.8	1.4	1.1	0.9	1.0	0.7	0.9	4.8
2qbr	1.2	1.8	1.9	1.4	2.9	1.3	1.0	5.0
1yat	0.2	0.5	0.5	0.5	0.5	0.4	0.5	5.1
1xp9	0.7	0.6	0.6	1.0	0.8	0.6	0.6	4.8
2hxm	1.2	1.4	1.1	1.4	1.1	0.5	0.7	5.1
2wca	0.4	0.6	0.7	0.7	0.4	0.4	0.4	7.5
2wi4	0.8	1.0	0.7	0.6	1.4	0.7	0.4	14.0
1uyg	2.5	2.5	2.3	2.1	0.8	1.9	2.6	5.0
2wi1	3.1	1.6	1.5	0.5	0.8	0.8	0.4	11.0
3fr2	0.7	1.2	0.8	0.8	1.5	0.8	0.7	3.3
3atm	0.2	0.4	0.4	0.3	0.6	0.2	0.5	2.3
5l7i	0.3	0.5	0.8	1.2	0.7	0.7	1.4	19.5

(bonded)									
6afa	1.000	1.000	1.000	1.000	1.000	1.000	1.000	1.000	0.999
(non-bonded)									
6pub	1.000	0.995	0.997	0.999	1.000	1.000	1.000	1.000	0.970
1owk	1.000	1.000	1.000	1.000	1.000	1.000	1.000	1.000	0.997
2qbr	1.000	1.000	1.000	1.000	0.999	1.000	1.000	1.000	0.997
1yat	1.000	1.000	1.000	1.000	1.000	1.000	1.000	1.000	0.997
1xp9	1.000	1.000	1.000	1.000	1.000	1.000	1.000	1.000	0.997
2hxm	1.000	1.000	1.000	1.000	1.000	1.000	1.000	1.000	0.997
2wca	1.000	1.000	1.000	1.000	1.000	1.000	1.000	1.000	0.994
2wi4	1.000	1.000	1.000	1.000	1.000	1.000	1.000	1.000	0.971
1uyg	0.999	0.999	0.999	0.999	1.000	1.000	0.999	0.999	0.996
2wi1	0.998	1.000	1.000	1.000	1.000	1.000	1.000	1.000	0.980
3fr2	1.000	1.000	1.000	1.000	1.000	1.000	1.000	1.000	0.998
3atm	1.000	1.000	1.000	1.000	1.000	1.000	1.000	1.000	0.999
5I7i	1.000	1.000	1.000	1.000	1.000	1.000	1.000	1.000	0.944

Comment 3: As reference for the geometry optimizations in the gas-phase, the authors chose the ω B97X functional. The reasoning that this functional was the one used to train the ANI-2x method is not convincing enough. The authors should soundly justify why this functional was selected as reference for the calculations. In addition, the used basis set was 6-31g(d), which lacks diffuse functions. Diffuse functions are often important to properly describe hydrogen bond interactions. The authors should check if changing the functional and basis set used for the optimizations affects the quality of the results in terms of agreement between the optimized geometries.

Our Response: Thank you for your helpful suggestion. To address your concerns, we have performed additional gas-phase and QR optimization of these 10 systems by adding the dispersion correction (i.e., ω B97X-D), using 6-31+G(d) basis set or changing to another popular and reliable DFT (M06-2X) method. As shown below, any of these changes in the method give almost same geometry (MAD (gas phase): 0.001 Å (bonds), 0.1-0.2° (angles) and 7-16° (torsions); MAD (QR): 0.001 Å (bonds), 0.1 ° (angles) and 0.2-0.3° (torsions)). Moreover, we have updated all results and analysis based on our latest optimization or QR calculations using ω B97X-D as the DFT method. The overall results and conclusions are not affected by using ω B97X-D. We

have added these new results and discussion in our revised manuscript and SI as follow:

Our Action:

Our revised manuscript:

Moreover, effects of dispersion correction, DFT functional and diffuse functions in the QM method on our QRs of the selected 10 drug/inhibitor (shown in Fig. 1) were examined. These selected 10 drug/inhibitor-protein structures refined based on M1 quantum refinements using ω B97X/6-31G(d), ω B97X-D/6-31+G(d) or M06-2X/6-31G(d) method as the QM method are very similar as those by ω B97X-D/6-31G(d) method (MAD: 0.001 Å, 0.0-0.1° and 0.2-0.3°, respectively; Supplementary Table 3).

Gas-phase geometry optimization. All drug/inhibitor molecules were also optimized by the above-mentioned DFT, MLPs and SE methods using Gaussian 16 as the geometry optimizer. In addition, effects of dispersion correction, DFT functional and diffuse functions were also examined using the selected 10 drug/inhibitor (shown in Fig. 1). These molecules optimized by ω B97X/6-31G(d), ω B97X-D/6-31+G(d) or M06-2X/6-31G(d) method are very similar as those by ω B97X-D/6-31G(d) method (MAD: 0.001 Å, 0.1-0.2° and 7-16°, respectively; Supplementary Fig. 2, and Supplementary Table 2). Therefore, ω B97X-D/6-31G(d) was used as the DFT method in this study.

Our revised Supplementary information 1:

Supplementary Table 25 MAD, RMSD and Correlation (R^2) of the optimized bond distances, angles, rotatable dihedrals for the selected 10 drugs/inhibitors (shown in Fig. 1) based on optimization in the gas phase using different functionals and basis sets compared to the DFT (ω B97X/6-31G(d)) method.

		ω B97X-D/6-31+G(d)	ω B97X-D/6-31G(d)	M06-2x/6-31G(d)
Bonds (Å)	MAD	0.001	0.001	0.001
	RMSD	0.004	0.003	0.004
	R^2	1.000	1.000	1.000
Angles (°)	MAD	0.2	0.1	0.2
	RMSD	0.4	0.3	0.3
	R^2	0.997	0.998	0.998
Dihedrals (°)	MAD	16	13	7
	RMSD	65	59	43
	R^2	0.661	0.718	0.846

Supplementary Figure 6 (a) Violin plot of derivations of the rotatable dihedrals (°) of the selected 10 drug/inhibitor (shown in Fig. 1) (b) Supposition of rivaroxaban and CPI-0610 (shows outliers of violin plot) and (c) Violin plot of derivations of the rotatable dihedrals (°) without rivaroxaban and CPI-0610 based on optimization in the gas phase using different functionals and basis sets compared to the DFT (ω B97X-D/6-31G(d)) method.

Supplementary Table 26 MAD, RMSD and Correlation (R^2) of the refined bond distances, angles, rotatable dihedrals for the selected 10 drugs/inhibitors (shown in Fig. 1) based on M1 quantum refinements using different functionals and basis sets compared to the DFT (ω B97X/6-31G(d)) method.

		ω B97X-D/6-31+G(d)	ω B97X-D/6-31G(d)	M06-2x/6-31G(d)
Bonds (Å)	MAD	0.001	0.000	0.001
	RMSD	0.002	0.002	0.002
	R^2	1.000	1.000	1.000
Angles (°)	MAD	0.1	0.0	0.1
	RMSD	0.2	0.1	0.2
	R^2	0.999	1.000	0.999
Dihedrals (°)	MAD	0.3	0.0	0.2
	RMSD	0.6	0.0	0.5
	R^2	1.000	1.000	1.000

Comment 4: In line 294, the authors state that: “Moreover, the RMSDs for the charged drugs were found to be larger than those for the neutral drugs when MLPs were used...”. This is, however, not the case for the AIQM1 MLP method (according to Figure 6), where the RMSD is larger for the neutral species compared to the charged ones. The authors should discuss this result.

Our Response: Thank you for your comment. We have updated the RMSD figure based on the new QR50 dataset (using ω B97X-D) results and modified the below discussion in our revised manuscript. As shown in the below Fig. 6 and Supplementary Table 36, structural flexibility of the large systems is another key factor in larger RMSDs, apart from the charge effect.

Our Action:

Our revised manuscript:

Moreover, apart from the charge effect, structural flexibility of the drug/inhibitor molecules is another key important factor in the larger RMSDs in gas phase (Supplementary Table 36).

(Updated) Fig. 6: RMSD of structures optimized using the ANI-2x, AIQM1, and GFN2-xTB methods compared to the ω B97X-D method for the neutral drugs/inhibitors in the gas phase (orange) and in the proteins after quantum refinement (blue) and for the charged drugs/inhibitors in the gas phase (red) and in the proteins after quantum refinements (green). The QR data by the **M7**, **M8**, **M10** and **M6R** schemes (ONIOM3(DFT:SE:MM), ONIOM3(ANI-2x:SE:MM), ONIOM3(AIQM1:SE:MM) and ONIOM2(SE:MM), respectively) were used.

Structural comparison in the gas phase and in proteins. The RMSDs of the optimized drug/inhibitor structures in the gas phase (0.40-0.74 Å for ANI-2x, AIQM1, and GFN2-xTB) were significantly higher than their corresponding structures in proteins refined by QRs (0.02-0.03 Å: **M8**, **M10**, and **M6R** relative to **M7**)..... Moreover, the RMSDs for the charged drugs/inhibitors were found to be slightly larger than those for the neutral drugs/inhibitors when ANI-2x were used, possibly due to the aforementioned limitation of ANI methods.the optimized drug/inhibitor structures in the gas phase (charged: 0.49 Å; neutral: 0.59 Å) than the ANI-2x method (charged: 0.74 Å; neutral: 0.67Å).

Our revised Supplementary Information 1:

Supplementary Table 27 RMSD of the all 50 drugs/inhibitors optimized in gas phase using ANI-2x, AIQM1 and GFN2-xTB compared to the DFT (ω B97X-D/6-31G(d)) method and RMSD of the all 50 drugs/inhibitors refined in the proteins using **M8**, **M10** and **M6R** with respect to **M7** as the reference (the charged systems are highlighted by red).

PDB ID	Gas phase			QR		
	ANI-2x	AIQM1	GFN2-xTB	M8	M10	M6R
1xbb	0.81	0.10	0.39	0.06	0.09	0.06
4hzz	0.40	0.22	0.12	0.04	0.02	0.02
5p9i	0.52	0.62	0.18	0.01	0.01	0.01
6jx4	0.78	0.20	0.23	0.03	0.02	0.05
2w26	0.23	1.05	0.10	0.02	0.03	0.04
4kra	0.90	0.98	0.60	0.05	0.03	0.03
4p6w	0.15	0.20	0.13	0.02	0.02	0.03
5hls	0.22	0.21	0.08	0.02	0.01	0.03
5kr1	0.44	0.26	0.26	0.01	0.01	0.02
7rfw	0.74	1.43	1.71	0.02	0.02	0.04
(bonded)						
7rfw	0.35	0.96	0.54	0.03	0.02	0.07
(non-bonded)						
1ajx	0.34	0.17	0.16	0.01	0.01	0.01
1br5	0.88	0.55	0.91	0.03	0.03	0.04
1d4i	2.01	1.50	0.42	0.01	0.01	0.01
1f9g	0.08	0.03	0.07	0.03	0.02	0.02
1lf3	1.46	0.54	0.39	0.01	0.02	0.03
1m7q	0.18	0.18	0.29	0.01	0.02	0.01
1nj1	0.32	0.39	0.56	0.04	0.04	0.03
1nox	0.27	0.17	0.16	0.01	0.01	0.01
1owk	0.31	0.08	0.08	0.03	0.01	0.02
1p1o	0.17	0.42	0.50	0.03	0.01	0.02
1pa9	0.81	0.17	0.16	0.03	0.03	0.02
1pf7	1.22	0.24	0.11	0.02	0.01	0.02
1uy9	0.14	0.26	0.10	0.01	0.01	0.01
1uyg	1.28	1.16	1.18	0.02	0.02	0.03
1xoz	0.41	0.36	0.78	0.01	0.01	0.01
1xp9	1.56	0.21	0.22	0.02	0.01	0.01
1yat	1.43	1.44	0.98	0.01	0.01	0.01
2a3i	0.12	0.08	0.07	0.01	0.01	0.02
2ax6	0.06	0.07	0.04	0.01	0.01	0.01
2hxm	0.40	0.15	0.23	0.03	0.01	0.02
2iwu	0.27	0.27	0.26	0.04	0.04	0.04
2p7z	0.53	0.64	0.16	0.08	0.02	0.03
2qbr	1.97	1.33	1.42	0.04	0.03	0.02
2wca	0.68	0.70	0.39	0.01	0.01	0.01

2wi1	0.52	0.15	0.15	0.01	0.01	0.01
2wi4	0.25	0.20	0.33	0.01	0.01	0.02
3atm	0.03	0.01	0.02	0.01	0.00	0.01
3eq7	0.49	0.25	0.22	0.01	0.01	0.01
3fr2	0.26	0.49	0.58	0.02	0.01	0.02
3qqa	3.34	3.25	0.88	0.01	0.01	0.01
4xh6	0.45	0.46	0.28	0.02	0.01	0.01
4yhm	1.03	0.13	0.53	0.03	0.01	0.02
4zb8	0.90	0.46	0.18	0.04	0.02	0.03
5ct1	0.85	0.18	0.28	0.02	0.03	0.02
5y1y	0.15	0.02	0.02	0.02	0.01	0.01
5y62	1.03	0.98	1.23	0.01	0.03	0.02
6pub	0.05	0.02	0.10	0.03	0.01	0.02
6v83	1.57	1.42	1.41	0.08	0.08	0.04
5l7i	1.60	1.81	1.80	0.02	0.01	0.03
6afa						
(bonded)	0.44	0.53	0.55	0.03	0.03	0.04
6afa						
(non-bonded)	1.34	0.21	0.20	0.03	0.03	0.05

Comment 5: The analysis of the efficiency of the method based on the CPU core-hours employed must be extended to all systems studied and not only to the two systems presented in Table 2.

Our Response: Thank you for your useful comment. As some QR simulations were performed on different CPU nodes, we have re-done some of our QR simulations on these 10 systems on the same CPU nodes to get more fair comparison. We have updated our analysis on the computational costs in our revised Table 2.

Our Action:

Our revised manuscript:

Updated Table 3. Computational cost (CPU core-hours) of different quantum refinements for the ten selected systems shown in Fig. 1. All calculations are performed on an Intel Xeon Gold 5218 CPU.

System	M1	M3	M5	M6	M3→M1
Imatinib	144.0	1.9	2.4	1.5	54.8
Rivaroxaban	188.4	53.1	50.9	15.0	186.1
Oseltamivir	1004.0	56.9	60.8	47.7	232.6
Ciprofloxacin	5987.6	350.8	426.9	323.2	2994.1

Mometasone	1328.1	162.2	1061.6	153.9	1188.1
CPI-0610	168.8	3.2	204.7	23.1	129.4
Darunavir	185.8	158.5	67.1	73.0	117.6
Ibrutinib	541.4	171.8	401.0	46.8	253.0
Osimertinib	5798.5	812.3	265.5	198.7	196.3
Nirmatrelvir	3948.4	6.7	10.3	31.5	626.8

Comment 6: The nomenclature used by the authors to refer to each of the methods in the text is difficult to follow if the reader is not constantly looking at Table 1. This issue could be ameliorated if the method is explicitly mentioned more often in the text. Even if this would slightly increase the length of the manuscript, it will definitely improve its readability.

Our Response: Thanks for your good feedback. We have followed your suggestion by mentioning the method more in our revised manuscript. Please see our highlighted revised manuscript file for our all additions.

Comment 7: The manuscript could be published in Nature Communications after major revisions, especially after the extension of the dataset mentioned above.

Our Response: We appreciate your effort and all suggestion, which considerably improves the quality of this work. We have tried our best to address your comments, especially with the extension of our datasets. We hope you find our latest changes acceptable or satisfactory.

Response to Reviewer #2

Comments: I co-reviewed this manuscript with one of the reviewers who provided the listed reports. This is part of the Nature Communications initiative to facilitate training in peer review and to provide appropriate recognition for Early Career Researchers who co-review manuscripts.

Our Response: Thank you so much for your positive comments and valuable suggestions. We have addressed your comments by performing more additional calculations and analysis. Please see our detailed response to the Reviewers 1 and 3.

Response to Reviewer #3

Comments: In this manuscript, Yan et al. report their computational development of multi-scale quantum refinement method. This method aims to improve or correct X-ray crystal structures of ten common drug molecules within the corresponding target protein structures by replacing robust machine learning potentials (MLPs; mainly ANI- or AIQM1-type) with more expensive yet reliable QM methods such as DFT or high-level CCSD(T) method, for the first time. Moreover, their combination of two different MLPs levels within the substrative ONIOM framework (i.e. ONIOM3(MLP1:MLP2:MM) and ONIOM4(MLP1:MLP2:SE:MM)) is novel and could inspire other computational chemists to adopt similar approach for other chemical and biochemical studies. Their quantum refinement results indicate that the proposed approach can give reliable and highly efficient results comparable to QM-level accuracy, especially the computationally prohibitive CCSD(T) method. A significant finding from this study is their computational evidence of the coexistence of bonded and nonbonded forms of the famous nirmatrelvir drug in the SARS-CoV-2 protein: revision/correction of this drug-protein structure. As SARS-CoV-2 is still a global issue and has attracted tremendous attention in different areas and communities (beyond just chemistry and biochemistry), this revised/corrected nirmatrelvir-SARS-CoV-2 protein structure should be of broad interest to readers of Nat. Commun. and other communities. Overall, this study likely advances biological structural refinement method and multi-scale simulation methods. I recommend the manuscript to be published in Nature Communications after a minor revision as follows,

Our Response: Thank you so much for your effort to review our work, positive comments and all constructive suggestion, which help us considerably improve our work.

Comment 1: It would be essential to give some more discussion on the limitations of MLPs. Is it possible to test one or two other MLPs besides the ANI and AIQM1 types?

Our Response: Thanks a lot for your useful comments. First, we have performed additional optimization calculations by using another recent MLP method (QDpi) for drug molecules (JCTC 2023, 1261). This MLP gives similar accuracy of our 10 systems. We have also added the below results, ref. 59, and more discussion about the limitations of MLPs in our revised manuscript and SI.

Our Action:

Our revised manuscript:

Due to the high dependence on training data, the MLPs are limited to few elements (e.g.: AIQM1: C, H, O, N; ANI-2x: C, H, O, N, F, Cl, S) or specify systems (ANI: neutral systems). To apply refinement of drug/inhibitor molecules containing more elements and to overcome such limitation while maintaining the highest accuracy on the core drug/inhibitor structures,

suggesting that the combination of several levels of MLPs via the ONIOM approach offers a promising avenue for **overcoming some limitations of MLPs and enhancing their advantages.**

Therefore, these results show that the combined MLP-CC with SE method can further resolve limitations of MLPs and be applied to much more elements and molecules.

To overcome the element restrictions in some MLPs, two different levels of MLPs were also combined for the first time by an extrapolative ONIOM approach and then applied as the unprecedented ONIOM3(MLP1:MLP2:MM) and ONIOM4(MLP1:MLP2:SE:MM) schemes for QR.

We are also optimistic that with the advances of more high-level and general MLPs developed by different groups,⁵⁵⁻⁵⁹

59. Zeng J., Tao Y., Giese T. J., York D. M. QD π : A Quantum Deep Potential Interaction Model for Drug Discovery. *J. Chem. Theory Comput.* **19**, 1261-1275 (2023).

Our revised Supplementary information 1:

Test on selected 10 drug/inhibitor (shown in Fig. 1)

Supplementary Figure 7 Violin plot of derivations of the refined bond distances (\AA), angles ($^\circ$), rotatable dihedrals ($^\circ$) of the selected 10 drug/inhibitor (shown in Fig. 1) structures optimized in the gas phase (neutral and charged groups), using QDpi, AIQM1, ANI-1ccx, ANI-2x, ANI-1x and GFN2-xTB methods compared to the DFT (ω B97X-D/6-31G(d)) method. ONIOM(MLP:ANI-2x) method was used for the molecules containing F, Cl and/or S elements, when AIQM1, QDpi, ANI-1ccx or ANI-1x was used.

Supplementary Table 28 Correlation (R^2) and MAD of the refined bond distances, angles, rotatable dihedrals for the selected 10 drug/inhibitor (shown in Fig. 1) based on optimization in the gas phase using AIQM1, ANI-1ccx, ANI-2x, ANI-1x and GFN2-xTB methods compared to the DFT (ω B97X-D/6-31G(d)) method. ONIOM(MLP:ANI-2x) method was used for the molecules containing F, Cl and/or S elements, when AIQM1, QDpi, ANI-1ccx or ANI-1x was used.

		QDpi	AIQM1	ANI-1ccx	ANI-1x	ANI-2x	GFN2-xTB
R^2	Bond	0.995	0.996	0.993	0.994	0.995	0.989
	Angle	0.981	0.986	0.960	0.959	0.960	0.980
	Rotatable Dihedral	0.182	0.345	0.297	0.435	0.333	0.655
MAD	Bond (Å)	0.004	0.004	0.006	0.005	0.004	0.008
	Angle (°)	0.64	0.57	0.92	0.85	0.84	0.73
	Rotatable Dihedral (°)	38.51	30.39	36.35	30.06	32.07	20.45

Supplementary Table 29 Correlation (R^2) and MAD of the refined bond distances, angles, rotatable dihedrals for neutral of the selected 10 drug/inhibitor (shown in Fig. 1) based on optimization in the gas phase using AIQM1, ANI-1ccx, ANI-2x, ANI-1x and GFN2-xTB methods compared to the DFT (ω B97X-D/6-31G(d)) method. ONIOM(MLP:ANI-2x) method was used for the molecules containing F, Cl and/or S elements, when AIQM1, QDpi, ANI-1ccx or ANI-1x was used.

		QDpi	AIQM1	ANI-1ccx	ANI-1x	ANI-2x	GFN2-xTB
R^2	Bond	0.997	0.997	0.995	0.996	0.997	0.991
	Angle	0.989	0.991	0.983	0.987	0.986	0.983
	Rotatable Dihedral	0.224	0.319	0.330	0.471	0.309	0.620
MAD	Bond (Å)	0.004	0.003	0.005	0.004	0.004	0.008
	Angle (°)	0.54	0.50	0.70	0.60	0.60	0.71
	Rotatable Dihedral (°)	35.26	31.07	33.22	26.84	31.16	21.63

Supplementary Table 30 Correlation (R^2) and MAD of the refined bond distances, angles, rotatable dihedrals for charged of the selected 10 drug/inhibitor (shown in Fig. 1) based on optimization in the gas phase using AIQM1, ANI-1ccx, ANI-2x, ANI-1x and GFN2-xTB methods compared to the DFT (ω B97X-D/6-31G(d)) method. ONIOM(MLP:ANI-2x) method was used for the molecules containing F, Cl and/or S elements, when AIQM1, QDpi, ANI-1ccx or ANI-1x was used.

		QDpi	AIQM1	ANI-1ccx	ANI-1x	ANI-2x	GFN2-xTB
R^2	Bond	0.987	0.986	0.979	0.980	0.980	0.979
	Angle	0.937	0.957	0.835	0.807	0.818	0.966
	Rotatable Dihedral	0.022	0.443	0.171	0.298	0.423	0.790
MAD	Bond (Å)	0.005	0.006	0.008	0.008	0.007	0.009
	Angle (°)	1.03	0.81	1.72	1.77	1.71	0.78
	Rotatable Dihedral (°)	51.16	27.74	48.53	42.55	35.65	15.85

Comment 2: The combination of two different MLP levels using the substrative ONIOM approach is interesting. Could such combination be applied to popular additive QM/MM method?

Our Response: Thank you for your insightful comments. It is possibly to combine two different MLP levels within an additive QM/MM framework. We have added the below discussion and new refs. 61-62 in our revised manuscript. The ref. 55 (being ref. 60 in this revised manuscript) was also updated.

Our Action:

Our revised manuscript:

As reported previously,⁶⁰⁻⁶² combination of two different MLP levels can readily be applied and extended to an additive QM/MM framework (eqs. 6-7).

$$E_{MLP-CC/MLP-DFT/MM} = E_{MLP-CC,model} + E_{MLP-DFT,intermediate} - E_{MLP-DFT,model} + E_{MM} + E_{QM-MM} \quad (6)$$

$$E_{MLP-CC/MLP-DFT/xTB/MM} = E_{MLP-CC,model} + E_{MLP-DFT,intermediate1} - E_{MLP-DFT,model} + E_{xTB,intermediate2} - E_{xTB,intermediate1} + E_{MM} + E_{QM-MM} \quad (7)$$

60. Chung L. W., *et al.* The ONIOM Method and Its Applications. *Chem. Rev.* **115**, 5678-5796 (2015).

61. Izsák R., *et al.* Quantum computing in pharma: A multilayer embedding approach for near future applications. *J. Comput. Chem.* **44**, 406-421 (2023).

62. Ojha A. A., Votapka L. W., Amaro R. E. QMrebind: incorporating quantum mechanical force field reparameterization at the ligand binding site for improved drug-target kinetics through milestoning simulations. *Chem. Sci.* **14**, 13159-13175 (2023).

Comment 3: The ω B97X functional with the 6-31G(d) basis set was used in this study. It is more reliable to test the effect of the empirical dispersion corrections, using the ω B97X-D functional, at least for the drug structures in gas phase.

Our Response: Thank you for your helpful suggestion. To address your concerns, we have performed additional gas-phase and QR optimization of these 10 systems by

adding the dispersion correction (i.e., ω B97X-D). As shown below, these new change in the method gives almost same geometry (MAD (gas): 0.001Å (bonds), 0.1-0.2° (angles) and 7-16° (torsions); MAD (QR): 0.001Å (bonds), 0.1 ° (angles) and 0.2-0.3° (torsions)). Moreover, we have updated all results and analysis based on our latest optimization or QR calculations using ω B97X-D as the DFT method. We have updated and added these new results and discussion in our revised manuscript and SI as follow:

Our Action:

Our revised manuscript:

Moreover, effects of dispersion correction, DFT functional and diffuse functions in the QM method on our QRs of the selected 10 drug/inhibitor (shown in Fig. 1) were examined. These selected 10 drug/inhibitor-protein structures refined based on M1 quantum refinements using ω B97X/6-31G(d), ω B97X-D/6-31+G(d) or M06-2X/6-31G(d) method as the QM method are very similar as those by ω B97X-D/6-31G(d) method (MAD: 0.001 Å, 0.0-0.1° and 0.2-0.3°, respectively; Supplementary Table 3).

Gas-phase geometry optimization. All drug/inhibitor molecules were also optimized by the above-mentioned DFT, MLPs and SE methods using Gaussian 16 as the geometry optimizer. In addition, effects of dispersion correction, DFT functional and diffuse functions were also examined using the selected 10 drug/inhibitor (shown in Fig. 1). These molecules optimized by ω B97X/6-31G(d), ω B97X-D/6-31+G(d) or M06-2X/6-31G(d) method are very similar as those by ω B97X-D/6-31G(d) method (MAD: 0.001 Å, 0.1-0.2° and 7-16°, respectively; Supplementary Fig. 2, and Supplementary Table 2). Therefore, ω B97X-D/6-31G(d) was used as the DFT method in this study.

Our revised Supplementary information 1:

Supplementary Table 31 MAD, RMSD and Correlation (R^2) of the optimized bond distances, angles, rotatable dihedrals for the selected 10 drugs/inhibitors (shown in Fig. 1) based on optimization in the gas phase using different functionals and basis sets compared to the DFT (ω B97X/6-31G(d)) method.

		ω B97X-D/6-31+G(d)	ω B97X-D/6-31G(d)	M06-2x/6-31G(d)
Bonds (Å)	MAD	0.001	0.001	0.001
	RMSD	0.004	0.003	0.004
	R^2	1.000	1.000	1.000
Angles (°)	MAD	0.2	0.1	0.2
	RMSD	0.4	0.3	0.3
	R^2	0.997	0.998	0.998
Dihedrals (°)	MAD	16	13	7
	RMSD	65	59	43
	R^2	0.661	0.718	0.846

Supplementary Figure 8 (a) Violin plot of derivations of the rotatable dihedrals ($^{\circ}$) of the selected 10 drug/inhibitor (shown in Fig. 1) (b) Supposition of rivaroxaban and CPI-0610 (shows outliers of violin plot) and (c) Violin plot of derivations of the rotatable dihedrals ($^{\circ}$) without rivaroxaban and CPI-0610 based on optimization in the gas phase using different functionals and basis sets compared to the DFT (ω B97X/6-31G(d)) method.

Supplementary Table 32 MAD, RMSD and Correlation (R^2) of the refined bond distances, angles, rotatable dihedrals for the selected 10 drugs/inhibitors (shown in Fig. 1) based on **M1** quantum refinements using different functionals and basis sets compared to the DFT (ω B97X/6-31G(d)) method.

		ω B97X-D/6-31+G(d)	ω B97X-D/6-31G(d)	M06-2x/6-31G(d)
Bonds (Å)	MAD	0.001	0.000	0.001
	RMSD	0.002	0.002	0.002
	R^2	1.000	1.000	1.000
Angles (°)	MAD	0.1	0.0	0.1
	RMSD	0.2	0.1	0.2
	R^2	0.999	1.000	0.999
Dihedrals (°)	MAD	0.3	0.0	0.2
	RMSD	0.6	0.0	0.5
	R^2	1.000	1.000	1.000

Comment 4: The authors discussed the computational costs in the ONIOM2 approach. Could the authors also compare the cost for the ONIOM3 method to provide a more comprehensive comparison?

Our Response: Thank you for your useful comment. As some simulations were performed on different CPU nodes, we have re-done some of our QR simulations for these 10 systems on the same CPU node to get more fair comparison. Due to the inability of MLP to parallelizing computations on CPU, it is hard to fairly compare ONIOM3(DFT:SE:MM) to ONIOM3(MLP:SE:MM). We do these computations using multiple cores. We have updated our analysis on the computational costs in our revised Table 3 and added Supplementary Table 37.

Our Action:

Our revised manuscript:

Table 3. Computational cost (CPU core-hours) of different quantum refinements for the ten selected systems shown in Fig. 1. All calculations are performed on an Intel Xeon Gold 5218 CPU.

System	M1	M3	M5	M6	M3→M1
Imatinib	144.0	1.9	2.4	1.5	54.8
Rivaroxaban	188.4	53.1	50.9	15.0	186.1
Oseltamivir	1004.0	56.9	60.8	47.7	232.6
Ciprofloxacin	5987.6	350.8	426.9	323.2	2994.1

Mometasone	1328.1	162.2	1061.6	153.9	1188.1
CPI-0610	168.8	3.2	204.7	23.1	129.4
Darunavir	185.8	158.5	67.1	73.0	117.6
Ibrutinib	541.4	171.8	401.0	46.8	253.0
Osimertinib	5798.5	812.3	265.5	198.7	196.3
Nirmatrelvir	3948.4	6.7	10.3	31.5	626.8

Our revised Supplementary information 1:

Supplementary Table 33 Computational cost (CPU core-hours) of different quantum refinements for different systems. All **M3** (sole MLP-based) computations were performed on a single CPU, while the and other calculations (**M1**, **M5-M10**) were performed on 12/24 Intel(R) Xeon(R) Platinum 9242 CPU.

	M1	M3	M5	M6	M7	M8	M10	M6R
3qqa	58.8	8.4	24.6	1.7	67.9	26.7	19.6	9.2
4xh6	56.2	1.7	5.7	18.5	85.0	55.8	48.6	28.3
4yhm	110.3	7.2	42.8	13.5	163.6	60.5	20.2	446.4
6v83	54.8	3.4	24.4	18.4	288.4	238.3	185.5	86.1
1ajx	17.2	12.3	11.7	10.6	791.0	264.3	530.7	55.5
1br5	65.1	0.9	6.3	24.1	93.0	7.8	11.2	3.3
1d4i	272.0	2.8	7.1	5.3	439.4	23.2	367.8	155.1
1f9g	142.6	15.2	82.6	87.8	363.6	42.4	90.5	110.9
1lf3	96.0	4.0	41.4	7.1	148.4	18.4	111.6	20.5
4zb8	64.8	4.9	32.6	6.4	98.2	30.8	40.3	3.8
1p1o	3.8	1.5	6.3	3.5	26.4	16.7	28.9	18.9
1pf7	23.7	2.7	43.9	10.7	11.3	0.2	12.3	0.2
1m7q	117.3	2.6	89.2	26.9	194.4	79.8	495.7	38.6
1pa9	25.7	0.8	6.7	11.0	16.6	26.9	31.2	38.1
1xoz	154.1	1.3	13.5	16.6	248.7	9.8	28.8	32.1
2ax6	105.7	2.5	4.7	6.0	13.1	27.7	33.6	12.3
2iwu	126.6	2.0	25.0	21.3	280.6	66.8	9.4	14.3
2p7z	24.4	0.6	25.7	1.1	79.3	13.1	48.7	111.6
1uy9	42.8	1.1	8.6	14.6	20.1	45.9	92.1	32.3
3eq7	22.3	18.2	10.5	43.8	57.6	34.4	34.3	62.7
5y1y	7.3	4.6	5.9	1.5	7.5	3.9	16.0	4.2
5y62	105.0	5.4	6.4	56.3	225.9	56.7	57.2	28.7
1nox	111.9	6.7	11.7	1.3	92.6	14.4	316.6	78.3
1nj1	57.1	2.0	28.2	88.5	843.2	29.0	24.1	285.6
2a3i	12.1	3.3	50.9	6.3	231.2	178.5	33.7	72.3
5ct1	9.0	4.1	28.8	37.1	5.3	4.7	29.7	43.3
6afa	6.3	2.0	2.9	3.0	21.6	8.2	7.5	14.2
(bonded)								
6afa	12.9	2.7	12.1	8.6	53.5	47.9	41.1	23.6

(non-bonded)								
6pub	101.9	11.7	17.8	20.4	190.1	64.5	41.8	40.2
1owk	28.9	2.0	10.2	3.7	161.0	48.6	58.2	29.7
2qbr	166.3	20.7	5.2	5.0	76.2	30.1	7.6	14.4
1yat	91.1	1.4	17.1	6.6	192.4	19.2	9.0	117.0
1xp9	477.9	5.2	18.4	70.2	319.9	827.6	57.8	80.8
2hxm	14.3	2.9	6.6	4.8	17.5	6.5	18.1	10.0
2wca	42.8	3.6	14.8	11.8	27.2	25.4	9.0	18.4
2wi4	23.5	4.8	9.8	2.6	78.2	14.4	10.4	16.5
1uyg	4.7	4.6	7.3	17.8	197.6	20.0	20.3	84.6
2wi1	22.5	11.0	11.1	1.8	10.5	9.4	20.1	40.8
3fr2	98.9	5.1	9.9	3.5	118.1	6.6	49.9	9.4
3atm	21.8	1.4	72.4	7.8	4.8	12.6	66.3	5.4
5l7i	13.7	86.3	61.5	14.2	184.7	255.3	176.8	131.7

Comment 5: SARS-CoV-2 is a virus name. The authors should give its protein name instead. Could the authors check any other available crystal structures of the same drug-protein that contain both the two bonded forms?

Our Response: We appreciate for your pointing our mistake as well as good suggestion. We have added “main protease” or “M^{Pro}” to specify the protein name in our revised manuscript and SI. Please see our highlighted revised manuscript file for our all additions and related changes. We have also checked other available crystal structures for the same system. We have added our below new analysis, ref. 52 and discussion in our revised manuscript and SI as follow:

Our Action:

Our revised manuscript:

Moreover, large RSZD score and electron density discrepancy of the key C-S bond linkage in another crystal structure of wild-type SARS-CoV-2 M^{Pro} (PDB ID: 7S19, Supplementary Table 42 and Supplementary Fig. 46), which may also imply the coexistence of the bonded and nonbonded forms. Furthermore, the coexistence of the bonded and nonbonded forms were recently proposed in another crystal structure of wild-type SARS-CoV-2 M^{Pro} (PDB ID: 7VH8).⁵²

52. Zhao Y., *et al.* Crystal structure of SARS-CoV-2 main protease in complex with

protease inhibitor PF-07321332. *Protein Cell* **13**, 689-693 (2022).

Our revised Supplementary information 1:

Supplementary Table 34 Reported available structures of wild-type SARS-CoV-2 main protease with nirmatrelvir (link to CYS). Values in parentheses of 7RFW are the results of our quantum refinement for the bonded form by **M7**.

PDB ID	Resolution (Å)	C _{drug-S_{cys}}	RSZD	
		Distance (Å)	Nirmatrelvir	CYS
7RFW ^a	1.73	1.922(1.812)	1.3(0.3)	6.5(0.2)
7RFS ^a	1.91	1.804	0.0	0.7
7SI9 ^b	2.00	1.805	2.1	2.5
7TE0 ^c	2.00	1.810	1.3	2.2
7VH8 ^d	1.59	1.814/4.251	2.3	2.2
7VLO ^e	2.02	A: 1.766	2.6	0.2
		B: 1.765	0.9	0.5
7VLP ^f	1.50	A: 1.765	1.5	1.2
		B: 1.763	2.6	0.8
7VLQ ^f	1.94	A: 1.766	0.2	0.3
		B: 1.767	0.0	0.5
8DZ2 ^g	2.13	A: 1.808	1.0	1.8
		B: 1.771	0.4	0.6

a. (2021) *Science* 374: 1586-1593; b. (2022) *Nat Commun* 13: 2268. c. (2022) *J Med Chem* 65: 8686-8698. d. (2022) *Protein Cell* 13: 689-693. (2023). e. (2022) *J Virol* 96: e0201321-e0201321. f. (2022) *J Virol* 96: e0201321-e0201321. g. *J Biol Chem* 299: 103004-103004

Supplementary Figure 9 Available crystal structures for nirmatrelvir (4WI) in wild-type SARS-CoV-2 main protease including the electron density maps (2mFo-DFc maps, contoured at 1.0 σ (blue), mFo-DFc maps, contoured at +3.0 σ (green), and mFo-DFc maps, contoured at -3.0 σ (red)).

Comment 6: I recommend to cite some related recent QM/MM methods combined with NLPs or NLPs for drugs, such as J. Phys. Chem. Lett. 2018, 9, 3232; J. Chem. Inf. Model. 2021, 61, 1066; Phys. Chem. Chem. Phys. 2022, 24, 1326; Phys. Chem. Chem. Phys. 2022, 24, 18559.

Our Response: We thanks for your suggested references. We have added them and one more ref. (refs. 55-59) in our revised manuscript. Please see below.

Our Action:

Our revised manuscript:

We are also optimistic that with the advances of more high-level and general MLPs developed by different groups,⁵⁵⁻⁵⁹

55. Wang H., Yang W. Force Field for Water Based on Neural Network. *J. Phys. Chem. Lett.* **9**, 3232-3240 (2018).
56. Liu Z., *et al.* Transferable Multilevel Attention Neural Network for Accurate Prediction of Quantum Chemistry Properties via Multitask Learning. *J. Chem. Inf. Model.* **61**, 1066-1082 (2021).
57. Liao K., Dong S., Cheng Z., Li W., Li S. Combined fragment-based machine learning force field with classical force field and its application in the NMR calculations of macromolecules in solutions. *Phys. Chem. Chem. Phys.* **24**, 18559-18567 (2022).
58. Cheng Z., Du J., Zhang L., Ma J., Li W., Li S. Building quantum mechanics quality force fields of proteins with the generalized energy-based fragmentation approach and machine learning. *Phys. Chem. Chem. Phys.* **24**, 1326-1337 (2022).
59. Zeng J., Tao Y., Giese T. J., York D. M. QD π : A Quantum Deep Potential Interaction Model for Drug Discovery. *J. Chem. Theory Comput.* **19**, 1261-1275 (2023).

REVIEWERS' COMMENTS

Reviewer #1 (Remarks to the Author):

The authors carried out extensive work and thoroughly addressed the points expressed by the reviewer. The updated manuscript is very much improved and I recommend it for publication in Nature Communications.

Reviewer #1 (Remarks on code availability):

The data was available under the provided link at the day of the review's submission.

Reviewer #2 (Remarks to the Author):

Reviewer #3 (Remarks to the Author):

In this revised manuscript, the authors have made improvements according to my suggestions. I recommend the manuscript to be published in Nature Communications.

Reviewer #3 (Remarks on code availability):

The code provides a useful README file and a detailed manual for running the applications. I could install and run the code correctly.